# Spatio-molecular domains identified in the mouse subthalamic nucleus and neighboring glutamatergic and GABAergic brain structures

Åsa Wallén-Mackenzie [1]✉, Sylvie Dumas [2,5], Maria Papathanou [1,5], Mihaela M. Martis Thiele[3], Bianca Vlcek [1], Niclas König [1] & Åsa K. Björklund [4]

The subthalamic nucleus (STN) is crucial for normal motor, limbic and associative function. STN dysregulation is correlated with several brain disorders, including Parkinson's disease and obsessive compulsive disorder (OCD), for which high-frequency stimulation of the STN is increasing as therapy. However, clinical progress is hampered by poor knowledge of the anatomical–functional organization of the STN. Today, experimental mouse genetics provides outstanding capacity for functional decoding, provided selective promoters are available. Here, we implemented single-nuclei RNA sequencing (snRNASeq) of the mouse STN followed through with histological analysis of 16 candidate genes of interest. Our results demonstrate that the mouse STN is composed of at least four spatio-molecularly defined domains, each distinguished by defined sets of promoter activities. Further, molecular profiles dissociate the STN from the adjoining para-STN (PSTN) and neighboring structures of the hypothalamus, mammillary nuclei and zona incerta. Enhanced knowledge of STN´s internal organization should prove useful towards genetics-based functional decoding of this clinically relevant brain structure.

[1] Department of Organismal Biology, Unit of Comparative Physiology, Uppsala University, SE-752 36 Uppsala, Sweden. [2] Oramacell, 75006 Paris, France. [3] National Bioinformatics Infrastructure Sweden, Science for Life Laboratory (NBIS), Division of Cell Biology, Department of Clinical and Experimental Medicine, Faculty of Medicine and Health Sciences, Linköping University, SE-581 83 Linköping, Sweden. [4] Department of Cell and Molecular Biology, National Bioinformatics Infrastructure Sweden (NBIS), Science for Life Laboratory, Uppsala University, SE-752 37 Uppsala, Sweden. [5] These authors contributed equally: Sylvie Dumas, Maria Papathanou ✉email: asa.mackenzie@ebc.uu.se

The subthalamic nucleus (STN) is a small excitatory brain structure of critical importance in motor, limbic and associative behavior[1]. Abnormal firing activity of STN neurons is correlated with Parkinson's disease (PD) and obsessive compulsive disorder (OCD), while degeneration of STN neurons is observed in supranuclear palsy and Huntington disease[2–5]. Subthalamotomy was long used as treatment of PD and today, deep brain stimulation (DBS) of the STN effectively relieves symptoms of advanced stage-PD and highly treatment-resistant OCD[4,6–8]. Further, STN-DBS is currently in trial for additional neurological and psychiatric conditions[9–11]. While clearly an important clinical target, the neurobiological underpinnings of the STN have remained elusive. One reason is poor knowledge of the internal organization of the STN. According to the prevailing tripartite model, the human STN is composed of three distinct domains, each subserving different roles in behavioral regulation. This model has been supported by anatomical[1,12–14], connectivity[15–17] and electrophysiological[18,19] studies in humans and non-human primates. However, the model does not fully explain the complexity of the STN in brain function, and is the focus of debates[20,21].

Solving the structural–functional organization of the STN should prove useful for understanding how distinct STN neurons subserve different roles, and how to achieve optimal DBS-targeting for treatment of various types of symptoms while avoiding side-effects. The rodent STN is often used to advance knowledge of STN neurocircuitry. Indeed, transgenics-based methodology, including optogenetics, provides the advantage of spatial and temporal precision. The STN is a glutamatergic structure expressing the *Vesicular glutamate transporter 2* gene (*Slc17a6/Vglut2*)[22]. STN neurons also express the *Pitx2* gene encoding a transcription factor which is crucial for their normal development[23,24]. Pitx2/Vglut2 overlap is complete and covers the extent of the STN[22]. Both *Vglut2* and *Pitx2* promoters have been implemented recently in optogenetics and conditional transgenics analyses for physiological and behavioral studies of STN neurocircuitry in mice[22,25–27]. To now advance precision within the STN structure, molecular patterns defining putative substructures would be useful. However, as in humans, the internal organization of the mouse STN has remained elusive.

To identify gene expression patterns within the subthalamic nucleus of mice, we implemented single-nuclei RNA sequencing (snRNASeq) of the *Pitx2*-positive STN followed through with histological analysis of 16 candidate genes of interest throughout the subthalamic area. Our results now demonstrate that the mouse STN is composed of at least four molecularly defined domains. Spatio-molecular organizational maps of the mouse STN were generated. The results contribute to the anatomical understanding of the STN, and should prove useful towards genetics-based functional decoding of this clinically relevant brain structure.

## Results

**Single nuclei RNA sequencing (snRNASeq) of Pitx2-Cre-positive cells identifies six gene clusters**. To address the STN at the genome-wide gene expression level, a transgenic mouse line carrying the enzyme Cre-recombinase under control of the *Pitx2*-promoter (Pitx2$^{Cre/+}$)[24] was crossed with the floxed mCherry-TRAP reporter line (Gt(ROSA)26Sor-mCherry-Rpl10a)[28] to generate mCherryTRAP$^{Pitx2-Cre}$ mice in which Pitx2-Cre-positive neurons could be identified by the fluorescent mCherry reporter (Fig. 1a). Analysis of brain sections at post-natal day 28 (P28) showed strong expression of the *mCherry* reporter gene in the STN and it was also detected in the associated para-STN (PSTN) as well as in the posterior and lateral hypothalamus (PH and LH,

respectively) (Fig. 1a). Accumulation of the tagged L10a protein in nucleoli allows for purification of nuclei via FACS after nuclear preparation[29]. To achieve quadruplicate samples for sorting, each of the two STN structures was dissected from four male mCherryTRAP$^{Pitx2-Cre}$ mice at age P28 and nuclei were extracted. To maintain independent samples, extracted nuclei were not pooled but single-sorted into two 384-well plates. Each plate contained 191 single nuclei from two mice and processed using Smart-seq2 protocol[30] for full-length single snRNAseq analysis (Fig. 1b and Supplementary Fig. 1). 138 nuclei did not fully meet the criteria set for quality control and were excluded from further analysis (Supplementary Fig. 2a–d).

To integrate data for grouping and identification of sources of heterogeneity, the Seurat R package was next employed[31,32]. This enabled the grouping of the remaining 630 samples into six clusters (Supplementary Fig. 3a). The number of detected genes was similar across the clusters, with the exception of the smaller yellow cluster, which contained nuclei with a higher number of detected genes (Supplementary Fig. 3b). Since signature markers for the yellow cluster were associated with the oligodendrocyte lineage (*Tmem125, Tmem63a, Cd9, Car14, Lpar1*), this cluster (*n* = 11) was excluded from further analysis (Supplementary Fig. 3c–i). The filtered dataset was subjected to a second round of unsupervised clustering. In this final Seurat analysis, 1946 variable genes were used and the remaining 619 nuclei again grouped into six clusters (clusters 0–5) with similar number of detected genes among the clusters (Fig. 1c–e). The median number of detected genes was 5379 and the median number of counts was 354,443 and was similar between the two plates and the four animals (Fig. 1f, g, Supplementary Data 3). Nuclei from both experimental plates and from all four biological replicates were represented in all clusters (Fig. 1h).

**Cluster analysis identifies a glutamatergic phenotype and allows characterization of gene expression profiles**. Next, Pitx2 was visualized in a t-SNE plot to verify that the dataset represented the expected population (Fig. 2a). Indeed, all clusters were *Pitx2*-positive and also shown to be of glutamatergic lineage marked by the strong expression of *Slc17a6* (*Vglut2*) and the sparse expression of GABA markers, such as the *Slc32a1*, encoding the Vesicular inhibitory amino acid transporter and *Gad1*, encoding Glutamic acid decarboxylase (Fig. 2a). In addition, other genes previously associated with the STN[24,33,34] such as *Lmx1a, Foxa1, Foxp1, Foxp2, Barhl1, Epha3* were expressed in most clusters, while *Lmx1b* was barely detected (Supplementary Fig. 4). In the blue cluster5, a few nuclei expressed *Foxp1* but none of the other known STN genes (Supplementary Fig. 4e). Differential expression analysis was run by the Model-based Analysis of Single-cell Transcriptomics (MAST) method (Fig. 2b and Supplementary Data 1). Each cluster was compared versus the rest using a p adjusted value of <0.01. Given the close nature of some of the clusters (Fig. 2b), MAST was also run between specific clusters for up- and downregulated genes (Supplementary Data 2).

Genes associated with the purple cluster0 were *Chrna4, Stac2, Fam19a1, Sds1, Bcan, Cntn4, Kcna1, Kcnab3* and *Fgf11* (Fig. 2c). However, they were not exclusive to this cluster, as *Kcnab3, Bcan, Kcna1* and *Cntn4* shared similar expression levels to the red cluster3 and blue cluster5, and to a lesser extent with the two green clusters (Fig. 2c and Supplementary Fig. 5a). MAST revealed *Kcnab3* and *Fgf11* as upregulated genes and *Tac1, Baiap3* and *Synpr* as downregulated genes in the purple cluster compared to the green cluster1 (Fig. 2c, d, Supplementary Data 2). Among the upregulated genes found in the green cluster1 were *Grik1, Synpr, Tac1, Ebf2, Ebf3, Cdh23* and *Penk* (Fig. 2d). About 50% of

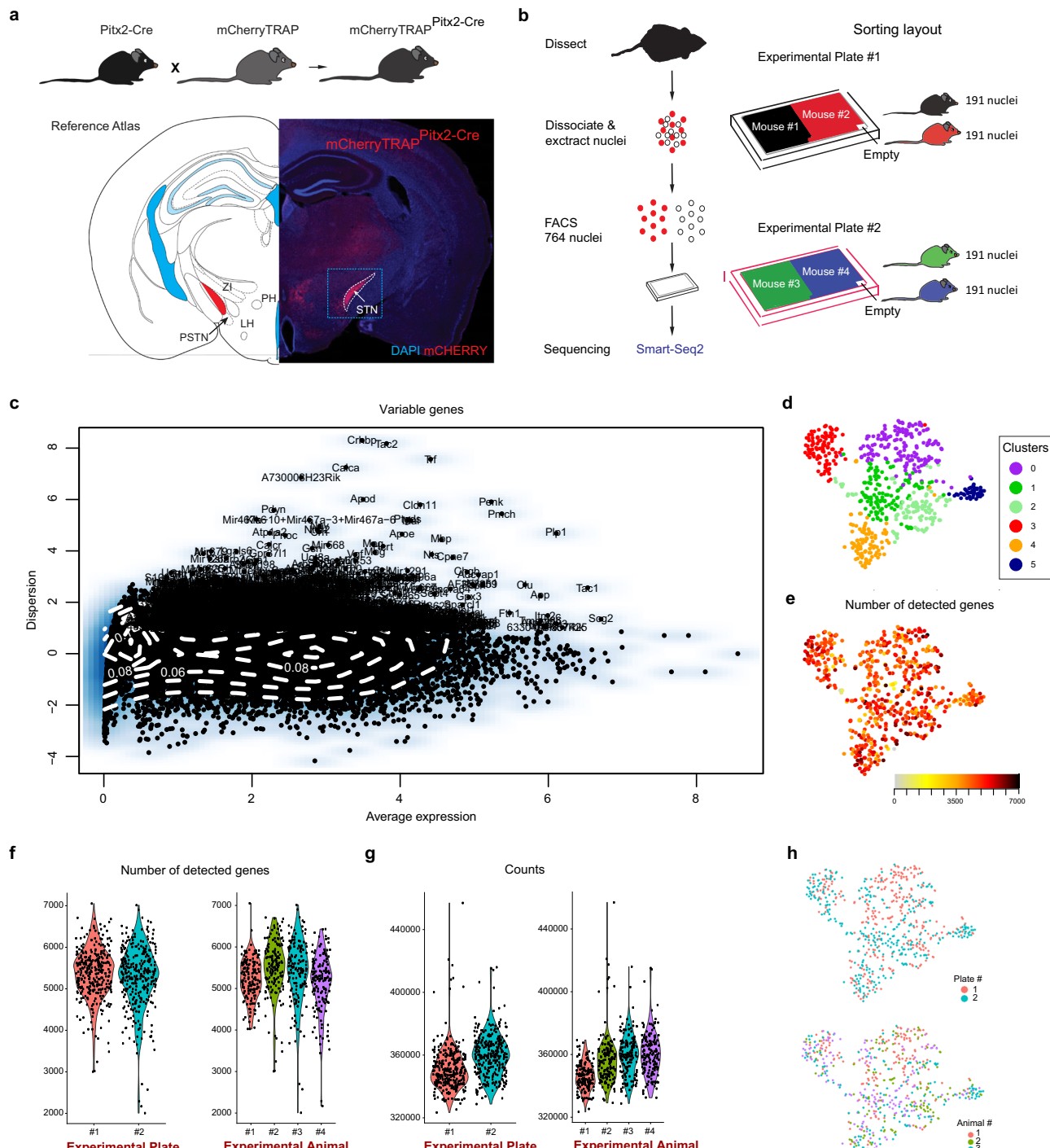

**Fig. 1 Single-nuclei RNA-sequencing (snRNASeq) of Pitx2-positive cells dissected from the subthalamic nucleus of Pitx2-Cre mice identifies six gene clusters. a** Schematic illustration of breeding strategy of Pitx2-Cre mice and mCherryTRAP reporter mice to generate mCherryTRAP^Pitx2-Cre mice. mCherry-fluorescence detected in the STN of mCherryTRAP^Pitx2-Cre mice; gross outline of dissected region shown in square (dotted lines). Also hypothalamic neurons in vicinity show mCherry fluorescence. Reference Atlas refers to illustration modified from published atlas;[54] red area corresponds to the STN. **b** Schematic illustration of experimental procedure from dissection to sequencing. **c** Identification of highly variable genes. The dispersion of the genes (variance/log-average expression) is plotted against the average expression (normalized log-expression, on *x*-axis). The minimum threshold to be considered highly variable was set at a dispersion of 1 on the *y*-axis. **d** t-SNE with six clusters (purple cluster0; green cluster1; light-green cluster2; red cluster3; orange cluster4 and blue cluster5). **e** Number of detected genes per nuclei (expression levels ranging from light gray (low) to black (high). **f** Violin plots showing the number of detected genes of each nuclei across the two experiment plates according to Seurat unsupervised clustering. **g** Violin plots showing the average number of transcripts across the two experiment plates. **h** tSNE-plots colored by experimental plate and by animal (biological replicate). Abbreviations: LH, Lateral hypothalamus; PSTN Parasubthalamic nucleus; PH, Posterior hypothalamus, STN: Subthalamic nucleus; ZI, Zona incerta Source data underlying these plots are shown in Supplementary Data 3.

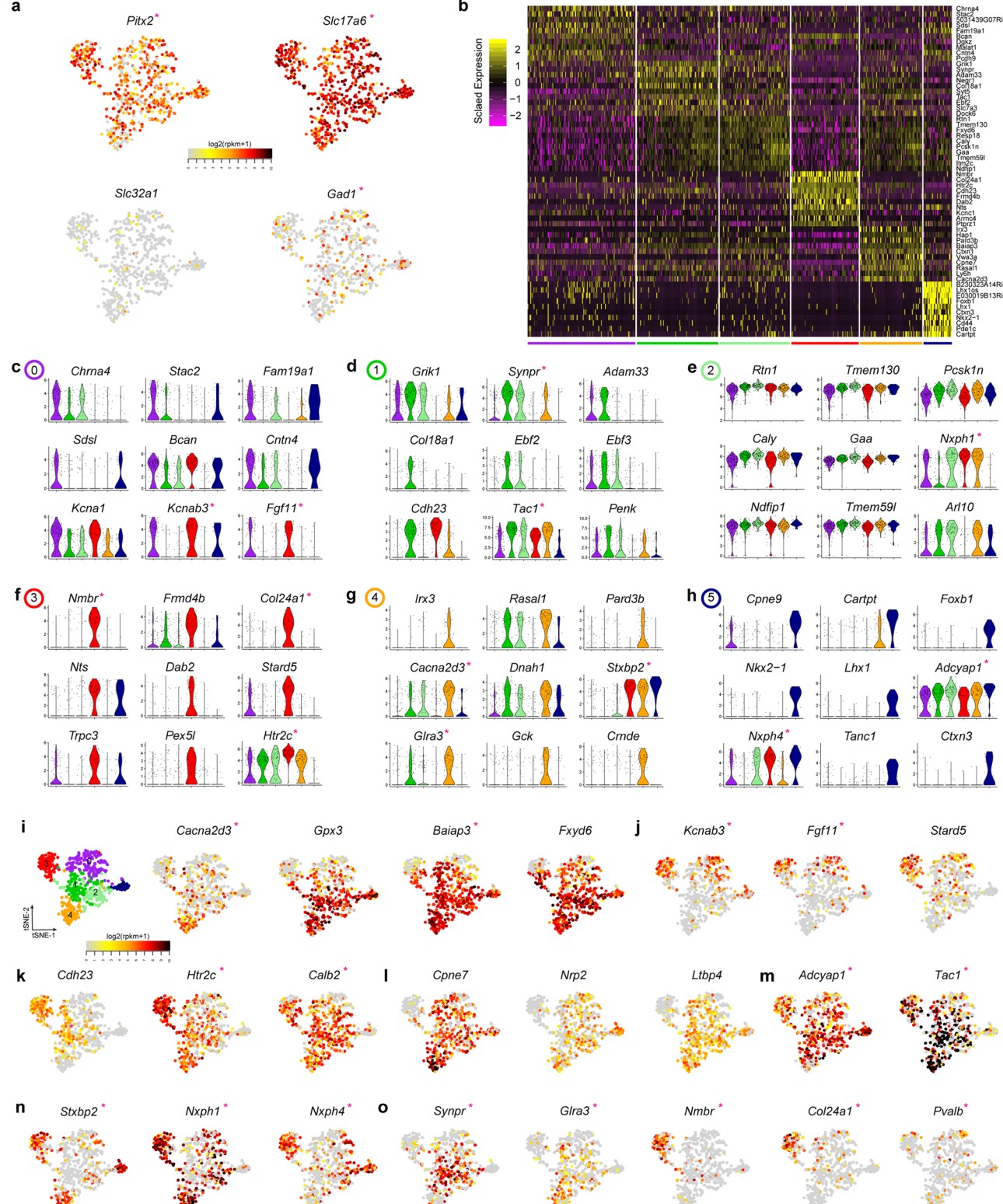

**Fig. 2 Bioinformatics characterization of the six Pitx2-positive clusters. a** Expression levels of *Pitx2, Slc17a6, Slc32a1 and Gad1* genes visualized in t-SNE plots. **b** Heatmap showing top ten genes differentially expressed for or characterizing each cluster. **c–h** Violin plots of nine selected markers of each population (purple cluster0; green cluster1; light-green cluster2; red cluster3; orange cluster4 and blue cluster5). **i, j** t-SNE plots with expression levels of *Cacna2d3, Gpx3, Baiap3 and Fxyd6 or Kcnab3, Fgf11 and Stard5* showing gradient expression along t-SNE axis-2. **k, l** t-SNE plots with expression levels of *Cdh23, Htr2c and Calb2 or Cpne7, Nrp2 and Ltbp4* showing distinct expression along t-SNE axis-1. **m** t-SNE of *Adycap1 and Tac1* showing ubiquitous expression across clusters. **n** t-SNE plots of *Stxbp2, Nxph1 and Nxph4* showing moderate to high expression in the red cluster3 and variable levels of in the remaining clusters. **o** t-SNE plots of genes showing unique expression among selective clusters (*Synpr, Glra3, Nmbr, Col24a1*) or expression in few nuclei in all clusters (*Pvalb*). Asteriscs indicate genes selected for mRNA analysis by fluorescent in situ hybridization (FISH). Source data underlying these plots are shown in Supplementary Data 3.

the nuclei in this cluster expressed these genes in moderate to high levels (Supplementary Fig 5b). Of these, *Grik1, Synpr, Ebf2, Ebf3* and *Penk* were also expressed in the light-green cluster2, a cluster that did not distinctly separate from cluster1 (Fig. 2e, Supplementary Fig. 5b, c). Comparison between the two green clusters showed *Cdh23* (higher in cluster1), *Nxph1, Nr4a2* and *Pcdh9* (higher in cluster2) to be the only differing genes between these clusters (Fig. 2d, e and Supplementary Data 2). The highest number of differentially expressed genes were seen in the red cluster3, which also showed high levels of *Pitx2* (Fig. 2f). In addition to *Kcnab3, Fgf11* and *Nxph1* (Fig. 2c, e) found in several clusters, numerous other genes, including *Nmbr* and *Col24a1* were almost exclusively expressed in the red cluster3 (Fig. 2f). Further, many genes were highly expressed in almost all neurons of the red cluster3, with partial expression in the other clusters, including *Stard5, Cdh23* and *Htr2c*, while *Nts* was predominantly expressed both in the red cluster3 and blue cluster5 (Fig. 2d, f and Supplementary Fig. 5d). Genes predominantly expressed in the orange cluster4 were *Irx3, Crnde* and *Glra3* in 25–50% of nuclei (Fig. 2g). Further, *Cacna2d3* and *Dnah1* were highly expressed in 75% of the nuclei of the orange cluster, and were also expressed in the green clusters (Fig. 2g, Supplementary Fig. 5e). Similarly, *Hap1* and *Baiap3* were strongly expressed in most nuclei of the orange and green clusters (Fig. 2b and Supplementary Fig. 5e). In contrast, *Stxbp2* and *Nxph1* were strongly detected in the orange cluster, but restricted in the green clusters (Fig. 2e, g). Finally, the blue cluster5 was positive for *Adcyap1, Stxbp2* and *Nxph4*, and further characterized by transcription factors including *Foxb1, Nkx2-1, Lhx1* (Fig. 2h and Supplementary Fig. 5f), several of which have been reported in the hypothalamus[35–37], while *Foxb1* has been described in the mammillary nucleus[38].

Many genes were thus expressed in more than one *Pitx2*-positive cluster while others appeared more selective, results that motivated further analysis. Next, by using the t-SNE algorithm to plot the distribution of the genes, a gradient expression from high to low along the t-SNE-2 axis was evident (Fig. 2i–o and Supplementary Fig. 6). This was apparent for several genes, such as *Baiap3*, which was highly expressed in several clusters (Fig. 2i), and for *Kcnab3, Fgf11* and *Stard5* that were predominantly expressed in the purple cluster0 and upper part of the red cluster3, but sparsely expressed in the other clusters (Fig. 2j). A difference was also found along the t-SNE-1 axis (Fig. 2k–o). For example, *Cdh23, Htr2c* and *Calb2* showed higher expression in the red cluster3 but low or absent expression in the blue cluster5 (Fig. 2k), while *Adcyap1* and *Tac1* were ubiquitously expressed (Fig. 2m). Additionally, *Stxbp2, Nxph1* and *Nxph4* were expressed in the red cluster3 and selectively detected in other clusters (Fig. 2n), while *Synpr, Glra3, Nmbr, Col24a1* and *Pvalb* showed more restricted or low expression within specific clusters (Fig. 2o).

Taken together, the results of the snRNASeq experiment argue both for the presence of *Pitx2*-positive neuronal populations positive for restricted combinations of genes, for gradients of multiple genes and also that several genes are found expressed in several clusters. This complexity motivated spatial analysis addressing gene expression patterns directly in mouse brain tissue.

## Spatial mapping of selected cluster genes identifies distinct patterns throughout the STN and neighboring structures.

Fluorescent *in situ* hybridization (FISH) was performed in multiple brain sections where each mRNA was first compared with Pitx2 mRNA. 16 genes from the six clusters were selected (listed in alphabetical order): *Adcyap1, Baiap3, Cacna2d3, Calb2, Col24a1, Fgf11, Glra3, Htr2c, Kcnab3, Nmbr, Nxph1, Nxph4,*

*Pvalb, Stxbp2, Synpr* and *Tac1*. For FISH analysis, mRNA-selective probes were synthesized (Supplementary Table 2; complete name of each mRNA, Supplementary Table 3) and analyzed in serial coronal sections encompassing the entire extent of the STN. P28 mice were used throughout to allow direct comparison with the results obtained in the snRNASeq experiment. In addition to the STN, surrounding structures that were identified to contain *Pitx2*-positive (Pitx2+) neurons based on *mCherry* reporter expression (Fig. 1a) were also analyzed to allow an unbiased histological approach: PSTN, PH, LH, retromammillary nuclei (RM) and medial mammillary nuclei (MM). Further, the zona incerta (ZI), was included in the analysis due to its anatomical position close to the dorsal edge of the STN.

Along with Pitx2 mRNA as control, also Vglut2 and Gad1 mRNAs were analyzed to allow identification of neurotransmitter phenotypes. Pitx2 mRNA was first verified in the STN with lower availability in the PSTN, PH, LH, MM, MR and its absence from the ZI (Fig. 3a, b). The STN and PSTN are directly associated anatomically but Pitx2 mRNA is lower in the PSTN than STN allowing a molecular distinction of the two structures (Fig. 3a). Analysis of all 16 selected mRNAs showed their co-localization with Pitx2 mRNA which validated their identification in the sequencing experiment (Table 1 and Fig. 3c–r). 11 of the 16 mRNAs selected from the snRNASeq were detected in the STN: Adcyap1, Calb2, Col24a1, Fgf11, Htr2c, Kcnab3, Nmbr, Nxph1, Nxph4, Pvalb, Stxbp2. However, 5 mRNAs were only found in adjacent structures: Baiap3, Cacna2d3, Glra3, Synpr and Tac1. Apart from the strongly Pitx2 mRNA-positive STN, it was apparent that several mRNAs were detected in both Pitx2+ and Pitx2- cells across some of these other structures (Table 1). FISH results from each structure are further described below. All data is summarized in Table 1.

*The STN*: Throughout the extent of the STN structure (Fig. 3a), Pitx2 mRNA co-localized to 100% with Htr2c and Kcnab3 mRNA (Fig. 3c, f). Htr2c was also identified in the ZI, PH and RM but was excluded from the MM and was very weak or absent in PSTN and LH (Fig. 3c–c′). Kcnab3 was equally strong in the STN, ZI and MM, excluded from the PSTN and modest in the other structures (Fig. 3f-f′). Htr2c and Kcnab3 mRNAs were found throughout the STN as was Nxph1 and Nxph4, the last two also strong in the PH (Fig. 3d-d', e-e′). Further, Stxbp2 and Calb2 mRNAs (Fig. 3h, m) were distributed over the STN area but less prominently so than Htr2c, Kcnab3, Nxph1 and Nxph4, and these mRNAs were also found in additional areas. Fgf11 and Nmbr showed a regional distribution with Nmbr more medial than Fgf11 (Fig. 3g, l). Nmbr was unique to the STN while Fgf11 was also in the ZI, however, both mRNAs were detected at very low levels. Pvalb mRNA, on the other hand, was strong in the central and lateral aspects of the STN while Col24a1 mRNA showed almost the opposite pattern with strong medial detection (Fig. 3j, k). Adcyap1 was sparse but detected in the central aspect of the STN while excluded medially and laterally (Fig. 3n). Regional distribution of several mRNAs along the STN structure thus points towards an internal organization of the STN with molecular properties that dissociate medial, central and lateral areas. All mRNAs detected in the STN co-localized with Pitx2 mRNA.

*The PSTN*: In contrast to the STN, the PSTN showed substantial amount of cells positive Baiap3, Glra3 and Tac1 mRNAs that were absent from the STN (Fig. 3i, o, q). These markers thus allow a spatio-molecular separation between the STN and PSTN beyond the substantially weaker expression of the *Pitx2* gene itself in PSTN than STN (Fig. 3a). Most of the PSTN neurons are Pitx2-negative (Fig. 3a–r), and Baiap3, Glra3, Tac1 were all found primarily in Pitx2- cells. In contrast to the more PSTN-restricted Tac1 and Glra3 mRNAs, Baiap3 was found

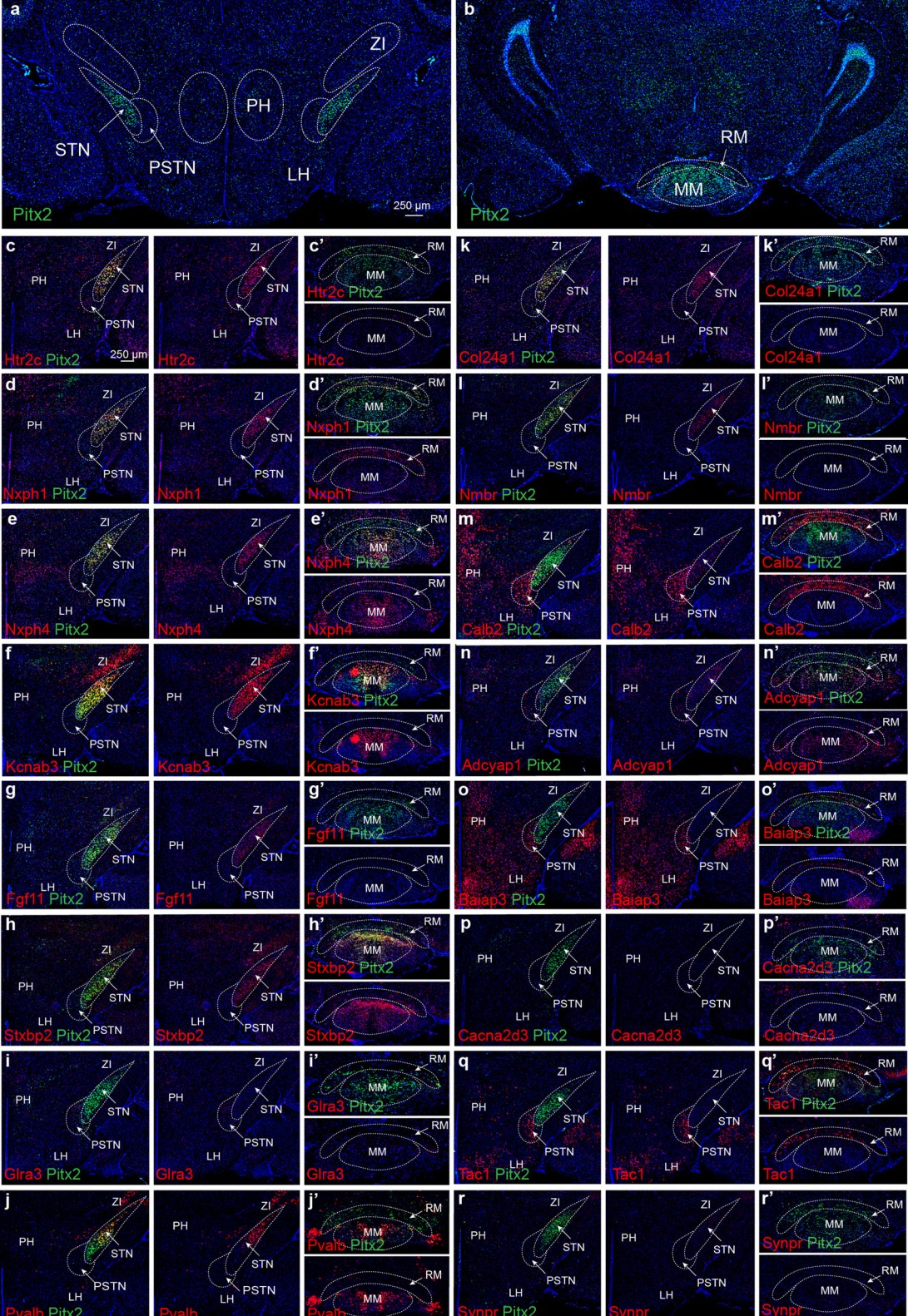

**Fig. 3 Fluorescent in situ hybridization (FISH) analysis of Pitx2 and 16 selected mRNAs representing cluster genes from the snRNASeq identifies unique patterns in the STN, PSTN and neighboring areas. a, b** Pitx2 mRNA primarily detected in the STN but also seen in the PSTN, LH, PH, ZI, MM and RM. **c–r** FISH analysis of Pitx2 mRNA (green) combined with 16 selected markers (red) within the STN, PSTN, ZI, LH and PH (in order presented: Htr2c; Nxph1; Nxph4; Kcnab3; Fgf11; Stxbp2; Glra3; Pvalb; Col24a1; Nmbr; Calb2; Adcyap1; Baiap3; Cacna2d3; Tac1; Synpr mRNAs). Regional distribution detected for all mRNAs in the ventral diencephalic areas assessed. All mRNAs found in the STN except Baiap3, Cacna2d3, Glra3, Tac1, Synpr. All mRNAs in the STN co-localize with Pitx2. Co-localization of mRNA with Pitx2 mRNA is shown in yellow. Scale bar 250 μm. Abbreviations: LH, Lateral hypothalamus; MM, Medial mammillary nucleus; PSTN Parasubthalamic nucleus; PH, Posterior hypothalamus, RM, Retromammillary nucleus; STN: Subthalamic nucleus; ZI, Zona incerta.

**Table 1 Summary of results obtained by expression analysis of 16 cluster genes with fluorescent in situ hybridization (FISH) analysis of mRNA on mouse brain tissue.**

| mRNA | STN | PSTN | PH | LH | MM | RM | ZI |
|---|---|---|---|---|---|---|---|
| Pitx2 | +++ | + | ++ | + | +++ | +++ | 0 |
| Vglut2 | +++ | ++ | +++ | +/++ | +++ | +++ | 0 |
| Gad1 | + | + | + | + | 0 | 0 | +++ |
| Adcyap1 | + | + | ++ | + | ++ | 0 | 0 |
| Baiap3 | 0 | +++ | +++ | +++ | 0 | + | + |
| Cacna2d3 | 0 | 0 | +/++ | 0 | 0 | 0 | 0 |
| Calb2 | ++ | +++ | +++ | ++ | 0 | ++ | + |
| Col24a1 | +/++ | + | + | + | 0 | 0 | 0 |
| Fgf11 | + | 0 | 0 | 0 | 0 | 0 | + |
| Glra3 | 0 | + | + | + | 0 | 0 | 0 |
| Htr2c | +++ | + | + | + | 0 | + | + |
| Kcnab3 | +++ | 0 | + | + | +++ | 0 | +++ |
| Nmbr | + | 0 | 0 | 0 | 0 | 0 | 0 |
| Nxph1 | +++ | + | +++ | + | 0 | +++ | 0 |
| Nxph4 | +++ | + | ++ | + | +++ | 0 | + |
| Pvalb | +/++ | + | + | + | ++ | 0 | ++ |
| Synpr | 0 | 0 | + | + | 0 | 0 | 0 |
| Stxbp2 | +/++ | 0 | + | 0 | +++ | + | + |
| Tac1 | 0 | +++ | + | + | 0 | +/++ | 0 |

Semi-quantitative analysis of degree of expression (graded as 0/+, +, ++, +++ for increasing density of positive cells). Vglut2 mRNA included as control for glutamatergic neurotransmitter phenotype. *PH* posterior hypothalamus, *LH* lateral hypothalamus, *MM* medial mammillary nucleus, *PSTN* parasubthalamic nucleus, *RM* retromammillary nucleus, *STN* subthalamic nucleus, *ZI* zona incerta.

broadly distributed in the hypothalamic areas and the MM, in both Pitx2-positive and Pitx2-negative areas, and also lining the STN in Pitx2-negative cells (Fig. 3o–o′). Substantial Calb2 mRNA was found in Pitx2+ and some Pitx2- cells of the PSTN (Fig. 3m). Nxph1, Htr2c, Col24a1 and Adcyap1 mRNAs were found in Pitx2+ and Pitx- cells at low levels in the PSTN (Fig. 3c, d, m, n), Nxph4 and Pvalb were detected at low levels exclusively in Pitx2- cells (Fig. 3d, j), and Cacna2d3 was not detected in either the PSTN or the STN (Fig. 3p). Summarizing the PSTN, based on the spatial expression data, primarily Tac1, but also Glra3, can be used to anatomically pinpoint the PSTN structure. Spatio-molecular mapping of the STN and PSTN thus demonstrate that these structures, despite their anatomical association, have different molecular profiles.

*The PH and LH*: Turning attention to the PH and LH, several of the mRNAs identified in the sequencing experiment were found distributed in these areas. In fact, the PH was positive for all mRNAs analyzed apart from Nmbr and Fgf11, while the LH in addition to these two mRNAs also lacked Cacna2d3 and Stxbp2. The differential expression of Cacna2d3 and Stxbp2 thus separate the PH and LH molecularly. Further, Cacna2d3 was exclusive in Pitx2-positive neurons of the PH out of all structures analyzed, suggesting its origin in the snRNASeq experiment be derived from PH neurons. Also Synpr was absent from the STN and all other structures but present in Pitx2-positive population of the PH (and Pitx2-negative population of LH). Adcyap1 was found in the Pitx2-positive population of the PH and Pitx2-negative neurons of the LH, but was not unique to the PH and LH as it was also found in the STN, PSTN, MM, in agreement with its ubiquitous presence in the cluster analysis. Nxph1, Nxph4, Calb2 and Baiap3 mRNAs were all readily detected in the PH (Fig. 3d, e, m, o) while Adcyap1, Cacna2d3 and Synpr were seen at low-to-moderate levels (Fig. 3n, p, r). Calb2 and Baiap3 mRNA labelings were strong in the adjoining LH (Fig. 3m, o). Calb2 and Synpr mRNAs were identified in both green sequencing clusters while Nxph1 differed between the two green clusters. Here, in the spatial expression analysis, Adcyap1, Synpr and Nxph4 were detected at lower levels in the LH than PH while Nxph1 mRNA was barely detected at all in the LH (Fig. 3d, e, n, r). In contrast, in the PH, Nxph1 mRNA labeling was strong and mostly confined to Pitx2-negative cells (Fig. 3d). Summarizing, in addition to Cacna2d3 and Stxbp2 mRNAs, also Nxph1 mRNA is thus segregated between the LH and PH.

*The ZI*: Most of the genes identified in the sequencing were absent or very sparsely detected in the ZI, which was identified as Pitx2-negative. However, Kcnab3 mRNA, which was prominent in the STN and MM, was also found in the ZI (Fig. 3f). Both Pvalb and Htr2c mRNAs showed regional distribution patterns within the ZI, thus representing ZI domains with distinct molecular identity (Fig. 3c, j).

*The MM and RM*: The MM and RM were the areas least positive areas for the selected markers, as only a minor subset of mRNAs could be detected here. Nmbr, Fgf11, Glra3, Synpr, Cacna2d3, Col24a1 were not represented by either RM or MM. However, visible differences between these anatomically adjacent areas were evident. Nxph1, Calb2 and Tac1 mRNAs were detected in moderate-to-high levels in the RM (Fig. 3d′, m′, q′) while Baiap3 and Htr2c were detected at low levels in this area (Fig. 3c′, o′). In contrast, none of these mRNAs were detected at all in the MM (same figures). Further, Kcnab3, Stxbp2, Nxph4, Adcyap1, Pvalb mRNAs were readily detected in the MM, but completely absent from the RM (Fig. 3e′, f′, h′, j′, n′). The molecular identity of neurons located in the MM and RM thus differ substantially. In fact, their spatio-molecular patterns are almost completely complementary. Pitx2 mRNA co-localized strongly with the identified markers in both the MM and RM with Nxph1 showing complete overlap with Pitx2 mRNA in the RM.

With the integrative data from both sequencing and FISH at hand, by summarizing the sequencing data in dotplot format (Fig. 4a), and by comparing expression levels detected in mouse brain sections (Fig. 4b), several genes identified by snRNASeq in the Pitx2-selected neuronal population can be confirmed as highly expressed in Pitx2+ neurons in the ventral diencephalon. As described, others are more poorly represented in brain sections than anticipated from the sequencing results. In summary, histological analysis allowed spatial mapping of all the 16 selected cluster genes originated from the snRNAseq experiment. The STN expressed 11 of these 16, while five were represented within the adjacently located PSTN, PH, LH, ZI, RM and/or MM. Apart from Nmbr which was only detected in the STN, none of the 11 genes was uniquely found in the STN. Instead all other genes expressed in the STN were also positive in at least one more its surrounding areas (Table 1, Fig. 3, Fig. 4b). However, within the STN structure, distinct expression patterns were discerned based on spatial location, abundance and co-localization with Pitx2 mRNA.

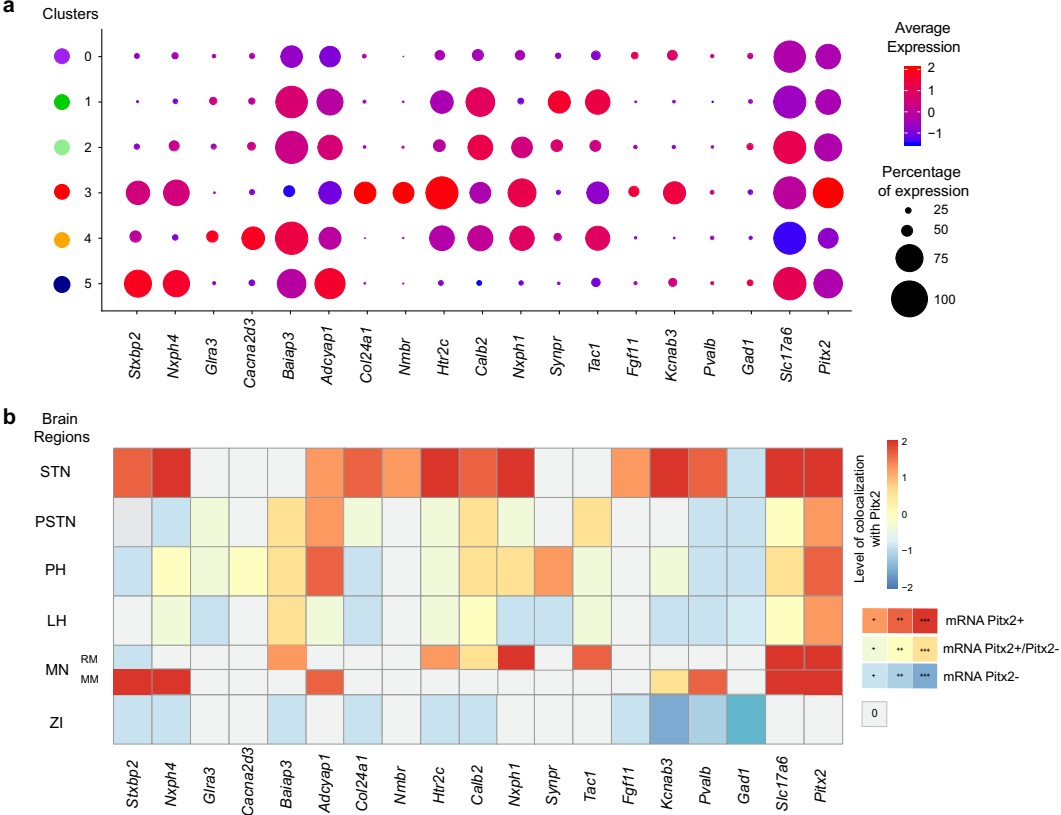

**Fig. 4 Summary of snRNA-seq and FISH data obtained with the 16 selected markers across the ventral diencephalon. a** Sequencing data summarized as dotplot for the selected markers across each cluster. Averaged expression shown from low (blue) to red (high) and size of dot representing the percentage of cells the gene. **b** Semi-quantitative analysis of FISH data summarized in a form of a heatmap (graded as 0, 0/+, +, ++, +++ for increasing density of co-localization with Pitx2 (shades of red) or detected in Pitx2-negative cells (blue)). Abbreviations: PH, Posterior hypothalamus; LH, Lateral hypothalamus; MN, Mammillary nucleus; MM, Medial mammillary nucleus; PSTN, Parasubthalamic nucleus; RM, Retromammillary nucleus; STN, Subthalamic nucleus; ZI, Zona incerta.

**Molecular distinction of four spatio-molecular domains within the STN structure.** Regional expression of certain genes within the mouse STN pointed towards the presence of molecularly defined domains within this structure. Combinatorial mRNA analysis was next performed on multiple sections across three anterior-posterior brain levels of the STN and its associated PSTN to outline the spatial relationship between mRNAs identified in these areas.

Based on the identification of Col24a1 and Pvalb mRNAs as differently distributed along the medio-lateral axis of the STN, their putative co-localization was of particular interest. Pitx2 mRNA, stronger in the STN than PSTN (Fig. 5a), outlined the entire STN/PSTN structure while Tac1 mRNA now enabled the PSTN to be completely dissociated from the STN (Fig. 5b). Again, Col24a1 mRNA was locally distributed within the medio-ventral aspect of the STN while barely detected in the remaining STN (Fig. 5c). On the other hand, Pvalb mRNA was most strongly detected in the dorso-lateral STN (Fig. 5d). Further, as described above, Pvalb was also found in a Pitx2-negative cell group dorso-laterally of the STN, located within the area of ZI (Fig. 5d, e). By comparing Pvalb with both Vglut2 (Fig. 5f) and Gad1 (Fig. 5g) mRNAs, it was clear that Pvalb+ neurons in the ZI are mostly GABAergic and Pvalb+ in the STN mostly glutamatergic, in accordance with the major neurotransmitter phenotypes of each area (Fig. 5f, g). Col24a1/Pvalb co-localization analysis further revealed that Col24a1 and Pvalb mRNAs were almost completely mutually exclusive within the STN, except for cells located in the intersection of their domains, that were positive for both (Fig. 5e). Col24a1 and Pvalb mRNAs thus enable classification of into three

distinct anatomical domains within the STN: the medial STNa positive for Col24a1, the central STNb positive for both Col24a and Pvalb which partly overlap, and the lateral STNc, positive for Pvalb (summarized in Fig. 5h).

Next, Htr2c, Kcnab3, Nxph1, Nxph4 and Calb2 mRNAs were compared in the STN, PSTN and ZI (Fig. 6a–d). As described above, these mRNAs were all present throughout the STN where they co-localized with Pitx2. In contrast, of these four mRNAs, only Calb2 was strong throughout the PSTN, while the other were sparse (Htr2c, Nxph1, Nxph4) or even absent (Kcnab3). Calb2 mRNA was dense in the PSTN and in the dorsal aspect of the medial STN (STNa) with only scattered distribution in the remaining STN (STNb, STNc). Further, Nxph1/Nxph4 co-localization analysis identified complete overlap throughout the STN apart from a dorsal strip which showed a Nxph1+/Nxph4− phenotype (Fig. 6c). Thus, while prominent detection of Htr2c, Kcnab3, Nxph1, Nxph4 and Pitx2 mRNAs characterizes the vast extent of the STN structure, a distinct dorsal domain lacks Nxph4. This domain, STNds (dorsal strip), was present primarily dorsally of the STNc. The spatio-molecular mapping thus enable the distinction of the mouse STN into four domains, STNa/b/c/ds, and also the clear molecular distinction of the STN from the PSTN structure (summarized in Fig. 5h, Fig. 6e, Table 2). In the ZI, Kcnab3 and Htr2c mRNAs were almost completely mutually exclusive (ventral vs dorsal ZI; Fig. 6a, b), while Calb2 mRNA was scattered in the dorsal part of the ZI (Fig. 6d). These findings thus allow further insight also in the GABAergic ZI, located at the dorsal border of the STN (schematics in Fig. 5h, Fig. 6e).

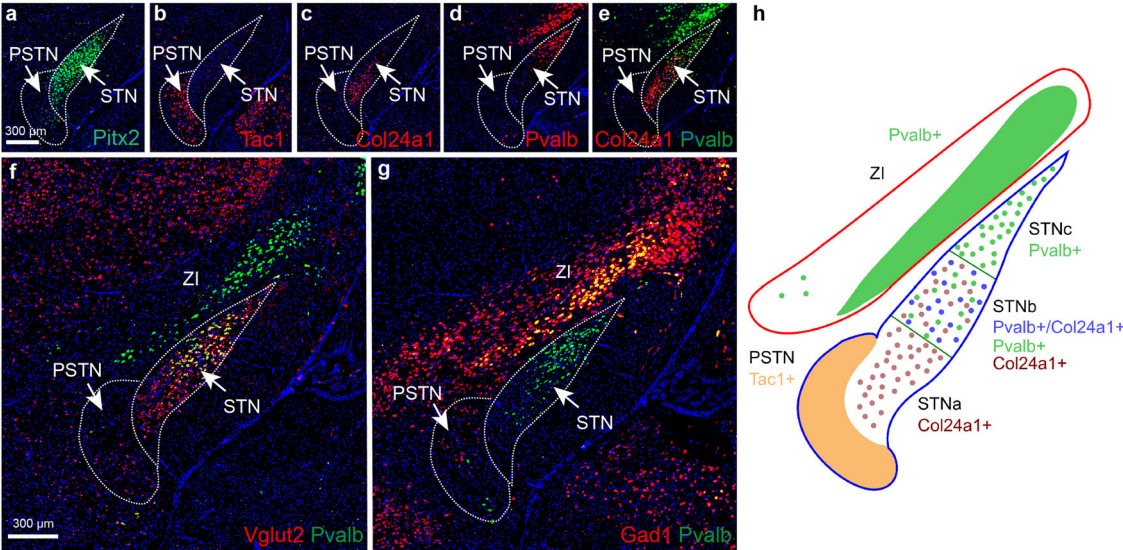

**Fig. 5 Fluorescent in situ hybridization (FISH) analysis identifies three major molecularly distinct domains across the STN and one domain within the ZI.** Regional ventral *vs* dorsal distribution of mRNAs. **a** Pitx2 mRNA covers STN and is less distributed in PSTN; **b** Tac1 in PSTN; **c** Col24a1 primarily in ventral STN; **d** Pvalb primarily in dorsal STN; **e** Col24a1/Pvalb2 co-localization identified in the central aspect of the STN. **f** Pvalb/Vglut2 co-localization in the STN; **g** Pvalb/Gad1 co-localization in the ZI but not in the STN. **h** Schematic illustration of results presented by FISH: PSTN defined by Tac1 mRNA; ventral STN by Col24a1 mRNA; central STN by co-localization Col24a1/Pvalb mRNAs; dorsal STN, and adjoining ventral aspect of the ZI, by Pvalb mRNA. Red outline of the ZI defining its GABAergic neurotransmitter phenotype (identified by Gad1); blue outline of the STN defining its glutamatergic neurotransmitter phenotype (identified by Vglut2). Scale bar 300 μm. Abbreviations: PSTN, Parasubthalamic nucleus; STN, Subthalamic nucleus; ZI, Zona incerta.

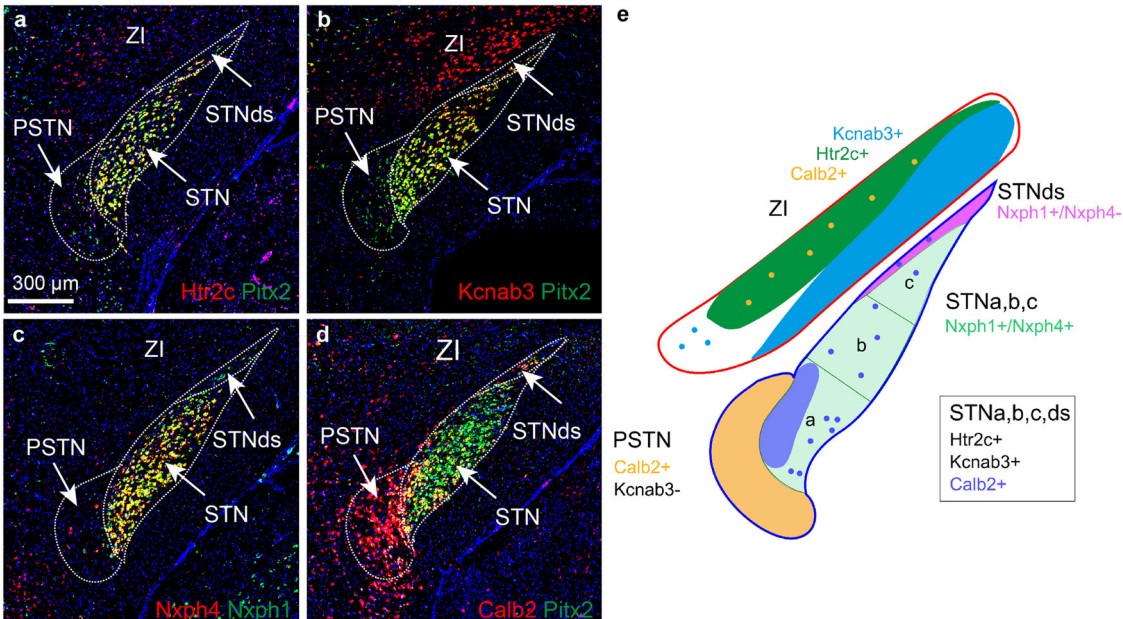

**Fig. 6 Identification of additional molecularly distinct domains in the dorsal aspect of the STN and within ZI.** Regional distribution of mRNAs: **a** Htr2c/Pitx2; **b** Kcnab3/Pitx2; **c** Nxph4/Nxph1; **d** Calb2/Pitx2 co-localiztion analysis; **e** Schematic illustration of results presented by FISH: PSTN defined by Calb2 and absence of Kcnab3, and STNa/b/c/ds by Htr2c, Kcnab3, Nxph1 and scattered Calb2. STNds is further distinct from the STNa/b/c (Nxph1+/Nxph4+) by absence of Nxph4 (Nxph1+/Nxph4−). Distinct patterns in ZI defined by either Knab3 (primarily ventral ZI), Htr2c (primarily dorsal ZI), Calb2 (scattered in dorsal ZI) mRNAs. Scale bar 300 μm. Abbreviations: PSTN, Parasubthalamic nucleus; STN, Subthalamic nucleus; STNds, Subthalamic nucleus, dorsal strip; ZI, Zona incerta.

Next, the following probe combinations were tested (Fig. 6a–r): (a) Pitx2/Vglut2; (b) Pitx2/Calb2; (c) Pitx2/Baiap3; (d) Htr2c/Baiap3; (e) Kcnab3/Baiap3; (f) Col24a1/Baiap3; (g) Pitx2/Glra3; (h) Pitx2/Tac1; (i) Htr2c/Tac1; (j) Pitx2/Gad1; (k) Pitx2/Htr2c; (l) Htr2c/Nxph1; (m) Htr2c/Nxph4; (n) Nxph1/ Nxph4; (o) Pitx2/ Col24a1; (p) Pvalb/Pitx2; (q) Pvalb2/Col24a1; (r) Tac1/Calb2. Results are presented in Fig. 7 and Table 2. By analyzing these combinations of mRNAs in multiple sections along the anteroposterior axis, additional patterns could be identified. Further, using these mRNAs as markers within the STN and

**Table 2 Summary of results obtained by expression analysis of 16 cluster genes with double-fluorescent in situ hybridization (FISH) analysis of mRNA in the subthalamic area.**

| | BRAIN REGION | | | | |
| | | STN | | | |
| mRNA | PSTN | STNds | STNa | STNb | STNc |
|---|---|---|---|---|---|
| Pitx2 | + | ++ | +++ | +++ | +++ |
| Vglut2 | ++ | ++ | +++ | +++ | +++ |
| Gad1 | + | + | 0 | 0 | + |
| Adcyap1 | + | 0 | 0 | + | 0 |
| Baiap3 | +++ | 0 | 0 | 0 | 0 |
| Cacna2d3 | 0 | 0 | 0 | 0 | 0 |
| Calb2 | +++ | + | ++ | + | + |
| Col24a1 | + | 0 | +++ | ++ | 0 |
| Fgf11 | 0 | 0 | 0 | + | + |
| Glra3 | + | 0 | 0 | 0 | 0 |
| Htr2c | + | ++ | +++ | +++ | +++ |
| Kcnab3 | 0 | ++ | +++ | +++ | +++ |
| Nmbr | 0 | 0 | + | + | 0 |
| Nxph1 | + | ++ | +++ | +++ | +++ |
| Nxph4 | + | 0 | +++ | +++ | +++ |
| Pvalb | 0 | ++ | 0 | ++ | +++ |
| Stxbp2 | 0 | 0 | + | + | + |
| Synpr | 0 | 0 | 0 | 0 | 0 |
| Tac1 | +++ | 0 | 0 | 0 | 0 |

Semi-quantitative expression analysis of degree of co-localization between mRNAs in the PSTN and four de novo identified domains of the STN: STNa, STNb, STNc and STNds (graded as 0, +, +, ++, +++ for increasing density of positive cells).
*PSTN* parasubthalamic nucleus, *STN* subthalamic nucleus, *ds* dorsal strip.

PSTN, it was possible to follow their anatomical distribution as their shape changed along the three-dimensional axes (S1–S3 section levels, throughout Fig. 7). Across the entire subthalamic area, Pitx2 and Vglut2 mRNAs were confirmed to show higher levels in the STN than PSTN, where both were sparse (Fig. 7a).

Throughout the S1-S3 levels, the whole extent of the STN was confirmed as positive for Pitx2, Vglut2, Htr2c, Nxph1 and Kcnab3 (Fig. 7a, d, e, f, k, l, m, n). Taken together with the co-localization analysis of each mRNA with Pitx2 shown above (Fig. 3), by comparing the patterns of several mRNAs (Fig. 7), it was evident that also additional markers followed the internal organization identified by Col24a1+(STNa), Col24a1+/Pvalb+(STNb) and Pvalb+(STNc) (Fig. 5 above). The following profiles could be pinpointed for each domain: STNa: Nxph4, Col24a1, Nmbr, Calb2, Stxbp2; STNb: Nxph4, Col24a1 (weaker than in STNa and partly overlapping with Pvalb), Pvalb, Stxbp2, Calb2, Nmbr, Fgf11, Adcyap1; STNc: Nxph4, Pvalb (stronger than in STNb), Fgf11, Stxbp2, Calb2; STNds: Pvalb2, Calb2, Gad1. Calb2 was strongest in STNa and scattered in STNb/c/ds. Further, in the STNds and STNc, Gad1 mRNAs was found in both Pitx2+ and Pitx2- cells. In addition to distinct markers representing each STN domain, the STNa/b/c/ds domains differed further in the actual extent of cells positive for each marker, and also their degree of overlap with Pitx2 and each other (Table 2, Figs. 5–7).

Finally, by summarizing all gene expression patterns identified in the STN and PSTN, our data allowed their visualization in three spatio-molecular maps across the anteroposterior axis (Fig. 8). Here, it is evident that the genes identified in the snRNASeq analysis and subsequently mRNA-mapped spatially in brain sections now enable the distinction of the STN from the anatomically associated PSTN, and also provide molecular

evidence for the subdivision of the STN structure into four internal domains (STNa/b/c/ds), the functional relevance of which remains to resolve.

## Discussion

The internal organization of the STN has been a matter of debate for many years with the prevailing tripartite hypothesis of the primate STN both supported and refuted[15,17,20,21]. Here, we identify 11 gene expression patterns that by defining four molecularly distinct domains directly support the concept of internal domains within the STN of the mouse. In addition, distinct mRNA patterns are shown to represent the PSTN, a structure anatomically associated with the medial aspect of the STN, the ZI which lines the dorsal aspect of the STN, as well as hypothalamic and mammillary areas in the vicinity of the STN.

The integrated snRNASeq—bioinformatics—histology approach allowed us to construct organizational maps of the subthalamic area by reaching from molecular analysis to in situ expression analysis of selected genes of interest. This approach also opened up for an inclusive analysis of several brain areas in the ventral diencephalon independent of their expression of the *Pitx2* gene, on which the initial selection of cells was based. Further, by anatomically addressing multiple sections across the area, by applying a range of combinations of probes, and by assessing neurotransmitter identity, several neurobiological aspects could be approached. This previously unidentified organization outlines the heterogeneity of the subthalamic area and opens up for opportunities towards increased understanding of the elusive STN and its neighboring structures.

By far the most striking findings were those related to the STN itself. Of the 11 STN mRNAs detected in the STN, Htr2c, Nxph1 and Kcnab3 were found at high levels throughout the STN forming a group of markers that, together with Pitx2 and Vglut2, can now be used to define the entire structure. In contrast, Nxph4, Col24a1, Pvalb, Stxbp2, Fgf11, Nmbr, Calb2 and Adcyap1 mRNAs showed spatially restricted distribution along the mediolateral and anteroposterior axes. Three major domains (STNa, STNb and STNc) along with one small domain (STNds) could be molecularly discerned by selective combinations of mRNAs, the four domains together forming the whole extent of the STN. Spatio-molecular mapping further showed that mRNAs distributed throughout the STNa/b/c domains were to the absolute majority positive for Pitx2. Since Pitx2 mRNA overlaps with Vglut2 mRNA, all these markers represent glutamatergic STN neurons, the main neurotransmitter phenotype of the STN[22,25]. In contrast, STNds was distinct from the STNa/b/c by expression of the *Pvalb* and *Calb2* genes in a mixture of Pitx2+ and Pitx2- cells. Further, a small group of cells in the STNc and STNds expressed the *Gad1* gene and were negative for Pitx2 mRNA, suggesting a local subpopulation of inhibitory neurotransmitter phenotype present within these STN domains.

The anatomical–functional organization of the STN is critical to fully grasp not least due to the strong clinical relevance of this brain nucleus. In the context of STN-DBS, it is important to bear in mind that surrounding structures are also critical to anatomically–functionally decipher, as the high-frequency stimulation exerted by DBS electrodes often will affect also these[39,40]. Further, in addition to the STN, the closely associated ZI is an important DBS target area. While the glutamatergic STN is a major target area for advanced-stage PD and highly-treatment resistant OCD, the highly GABAergic ZI structure is critical for alleviation of essential tremor[11]. Here we show that not only the STN, but also the ZI, contain domains that can be distinguished by molecular markers. Within the STN structure itself, DBS electrodes in PD are most commonly placed in the dorsolateral aspect,

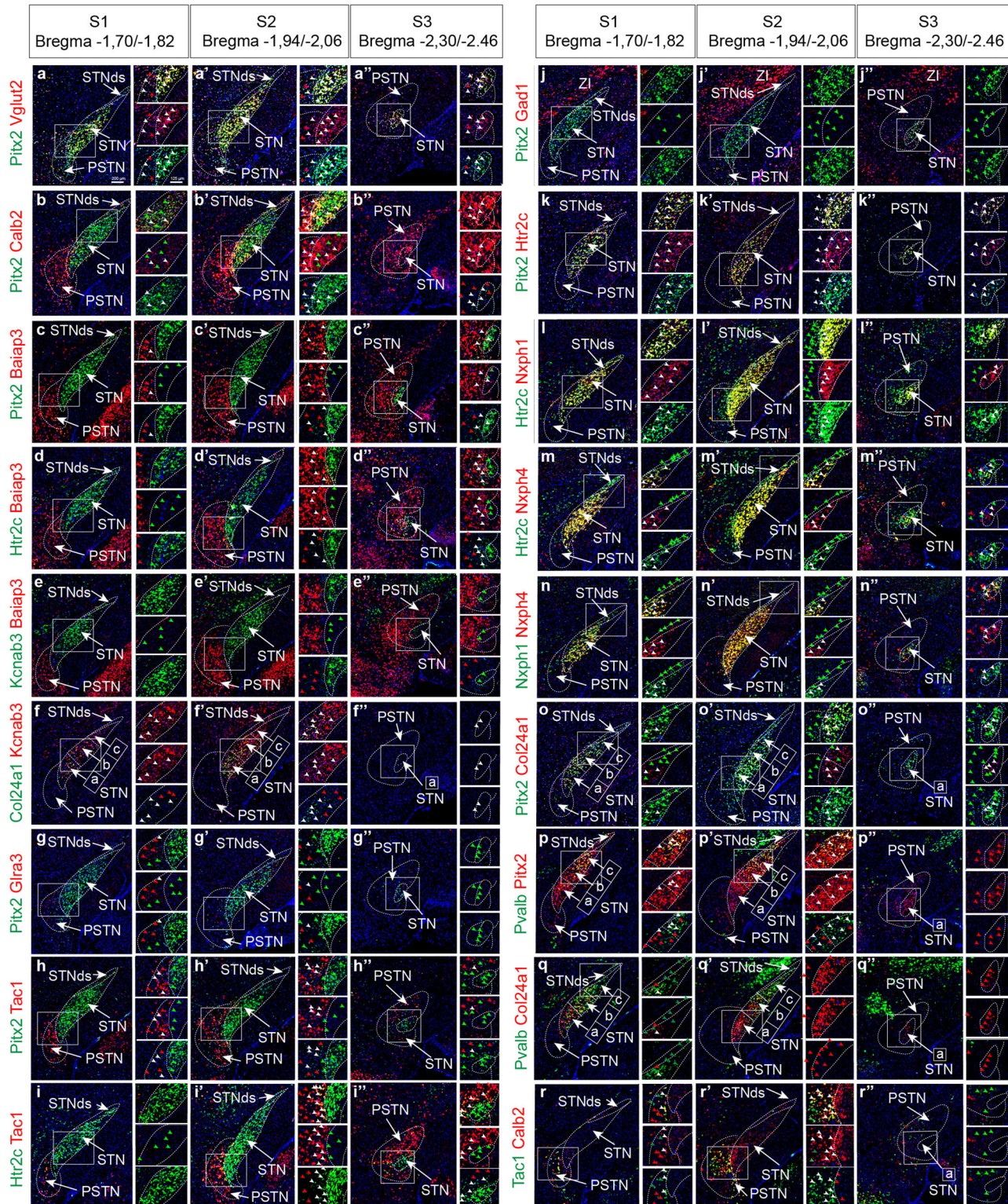

**Fig. 7 Altogether four distinct spatio-molecular domains identified within the mouse STN defined by combinatorial expression patterns.**
**a–r** Combinatorial FISH analysis comparing mRNAs towards each other allows distinction into STNa/STNb/STNc/STNds domains based on detection of several mRNAs. Three levels along the antero-posterior axis analyzed for complete revelation across the STN structure. mRNAs analyzed, in order presented: **a** Pitx2/Vglut2; **b** Pitx2/Calb2; **c** Pitx2/Baiap3; **d** Htr2c/Baiap3; **e** Kcnab3/Baiap3; **f** Col24a1/Kcnab3; **g** Pitx2/Glra3; **h** Pitx2/Tac1; **i** Htr2c/Tac1; **j** Pitx2/Gad1; **k** Pitx2/Htr2c; **l** Htr2c/Nxph1; **m** Htr2c/Nxph4; **n** Nxph1/Nxph4; **o** Pitx2/Col24a1; **p** Pvalb/Pitx2; **q** Pvalb/Col24a1; **r** Tac1/Calb2 presented in green/red with corresponding color of arrows and with white arrow heads representing co-localization. Scale bar 200 μm and 125 μm (insets). Abbreviations: dsSTN, dorsal strip of STN; FISH, fluorescent in situ hybridization; STN, Subthalamic nucleus; PSTN, Parasubthalamic nucleus; ZI, Zona incerta.

Spatio-molecular organization of the STN and PSTN

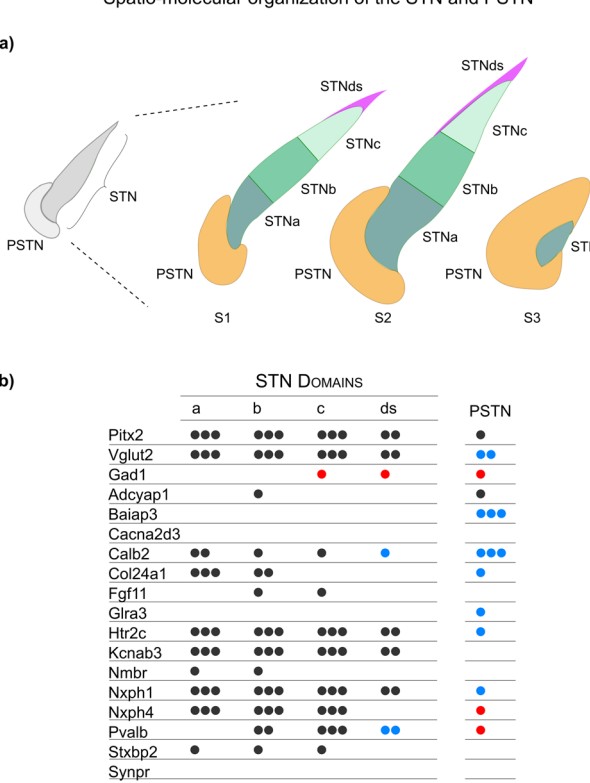

**Fig. 8 Summary of the spatio-molecular organization defining four internal STN domains, and their separation from the PSTN. a** Left: Gross schematics of the STN and PSTN in dark and light gray representing the previously known distribution of Pitx2 and Vglut2 mRNAs which are both stronger in STN than PSTN. Right: snRNASeq and FISH data presented in present study now allow spatio-molecular distinction of the STN and PSTN based on molecular markers and anatomical positions. mRNA pattern analysis at three brain levels (S1-S3) shows how the anatomy of the STN and PSTN changes with anteroposterior levels, with the STN surrounded by PSTN at the S3 level. The data also allows further separation of the STN structure into four molecularly distinct internal domains: the STNa, STNb, STNc and STNds. Note that STNds represents a dorsal strip only present at the S1 and S2 levels. At S3 level, only STNa and PSTN detected. **b** Table shows representation of mRNAs detected in each STN domain (STNa, STNb, STNc and STNds) and in the PSTN by FISH analysis in mouse brain tissue; data also presented in Table 2. Note that co-localization with Pitx2 mRNA is presented with color coding: Black mRNA, Pitx2+; Blue mRNA, Pitx2+/Pitx2-mixed; Red mRNA, Pitx2-. Abbreviations: FISH, fluorescent in situ hybridization; STN, Subthalamic nucleus; PSTN, Parasubthalamic nucleus.

corresponding to the sensorimotor domain according to the tripartite model, while in the limbic and/or associative domain for OCD treatment[6,39,41,42]. However, the efficiency varies substantially between individuals. Various side-effects have been reported, often argued to be a direct consequence of missing knowledge of underlying neurobiology[6,8,18,19]. Clearly, it is of major importance to find the optimal site for placement of electrodes within each target structure and for each type of symptom[43]. For this to be possible, functional studies in mice in which precise anatomical positions can be directly correlated with physiological and behavioral output have strong potential to contribute powerful advances, provided detailed spatio-molecular maps of the structural organization are available.

By identifying markers that define internal domains, the present study provides a highly useful toolbox of spatially precise promoter activities that can be functionally implemented in experimental mouse genetics. Using knockout strategies, we and others have shown that the STN is dependent on both Pitx2[23,24] and Vglut2[22] for normal development. Recently, we and others have used Pitx2-Cre or Vglut2-Cre mice for optogenetic activation or inhibition of the STN in adult mice[22,25,27]. Cre-driven viral-genetics in mice, including optogenetics, tract-tracing and electrophysiological approaches, has outstanding capacity to unravel neurobiological underpinnings that may help towards increased selectivity and reliability of clinical interventions[44]. However, the key to experimental selectivity and precision lies in the availability of spatially selective promoters for driving Cre-recombinase[26]. Here, our identification of promoter activity defining internal STN domains opens up a previously unrecognized dimension for anatomical–functional decoding of discrete populations of neurons within the STN. Using the previously unidentified promoters to drive selectivity in each internal STN domain will allow decoding of STN´s role in motor, limbic and associative functions at a higher level of resolution than previously possible. With such information, the four-domain structure identified by molecular profiles should be relevant to compare to the tripartite model of the primate STN. Further, combinatorial gene expression patterns can help towards solving complex functional roles using recent methodological advances of intersectional genetic approaches[45]. In this context, also the putative functional relevance of gradient expression can be deciphered. For example, Col24a1 and Pvalb gradients across the STN could be deciphered functionally to address the clinically relevant hypothesis of putative overlapping functional domains (associative/limbic and associative/motor) that may be critical in STN-DBS applications[16,46].

Mapping of Pitx2 mRNA itself showed substantially higher levels in the STN over PSTN, PH, LH, MM, RM at the stage of analysis (P28), and it was also established that Pitx2 is absent from the ZI. Interestingly, all these areas contained expression of subsets of genes found in the unsupervised cluster analysis. In fact, five of the cluster genes analyzed in the spatial mapping were not detected at all in the STN but were instead only expressed in these neighboring areas. These findings point towards the enrichment not only for STN neurons in the manual dissection of the STN followed by sorting for a Pitx2-Cre-driven reporter neurons. Primarily Tac1 and Glra3 mRNAs distinguish the PSTN from the STN with Tac1 as the most potent marker due to its high and almost selective detection in the PSTN. In the context of the hypothalamus, recent studies have explored the diversity of this heterogeneous area by means of single cell RNA-seq[35–37]. Here, additional genes distinguishing between hypothalamic neurons of the PH and LH were identified. In contrast to the abundance of cluster genes expressed in hypothalamic neurons, only few were found in the MM and RM. However, the RM and MM displayed almost completely complementary patterns of mRNAs selected from the STN-based cluster analysis. Also, the blue cluster showed unique expression of Foxb1 which defines the MM, but which is absent from the STN[38], suggesting the contribution of MM cells to this cluster.

This study addressed the molecular heterogeneity of the STN and additional Pitx2+ structures across the ventral diencephalon using an integrated transcriptomics-histological approach. Additional analyses will likely be required to fully outline the spatio-molecular composition of the STN. However, current data clearly point towards the presence of molecularly defined domains within the STN structure. The internal organization of STN domains identified here provides enhanced anatomical insight,

and also lay the grounds for advanced functional decoding of the complex STN structure in physiology and behavior.

## Methods

**Animal housing**. Mice of both sexes were used in the study. Mice were housed on a standard 12 h sleep/wake cycle (7:00 A.M. lights on, 7:00 P.M. lights off). Mice were provided with food and water ad libitum and housed according to Swedish legislation (Animal Welfare Act SFS 1998:56) and European Union legislation (Convention ETS 123 and Directive 2010/63/EU). All experiments were conducted with permission from the local Animal Ethical Committee (Uppsala Djurförsöksetiska Nämnd, Uppsala, Sweden).

**Generation and genotyping of transgenic mice**. Genotyping of mice was performed by PCR analysis (Supplementary Table 1). Pitx2$^{Cre/+}$ mice[24] were bred with the floxed mCherryTRAP reporter line (Gt(ROSA)26Sor-mCherry-Rpl10a)[28], generating mCherryTRAP$^{Pitx2-Cre}$ mice.

**Nuclear extraction and FACS sorting of mCherry-tagged nuclei**. Male mCherryTRAP$^{Pitx2-Cre}$ mice at postnatal day 28 (P28) were euthanized and brains quickly removed and put in PBS on ice. Each brain was first cut into 1mm-thick coronal section and the two subthalamic nuclei were dissected out and snap-frozen on dry ice. Tissue was thawed and dissociated using a 1-ml dounce homogenizer (Wheaton) in ice-cold lysis buffer (0.32 M sucrose, 5 mM CaCl$_2$, 3 mM MgAc, 0.1 mM Na$_2$EDTA, 10 mM Tris-HCl, pH 8.0, 1 mM DTT, and 1× complete proteinase inhibitor, 0.1 mM Na$_2$EDTA-free (Roche)). The homogenate was slowly added over a sucrose layer (1.8 M sucrose, 3 mM MgAc, 10 mM Tris-HCl, pH 8.0, and 1 mM DTT) before a centrifugation process of 2 h 20 min at 30,000× $g$ (Beckman J-25 centrifuge with a J13.1 rotor). The supernatant was carefully aspirated and the extracted nuclei in a form of a pellet were resuspended in a nuclear storage buffer (15% sucrose, 10 mM Tris-HCl, pH 7.2, 70 mM KCl, and 2 mM MgCl$_2$) containing an additional proteinase inhibitor (Complete, Roche) and an RNAse inhibitor 40 U/µL (RNAseOUT, Invitrogen). Resuspended nuclei were filtered through a 30-µm nylon cup filcon (BD Biosciences, 340625) into BSA-coated FACs tubes for sorting.

Fluorescence-Activated Cell Sorting (FACS) was performed using a BD Influx System and the BD FACS$^{TM}$ software (BD Biosciences) to isolate mCherry-tagged nuclei and single-sort into two 384 well plating containing 2.3 µl of ice-cold lysis buffer with external RNA controls consortium (ERCC) spike ins for subsequent quality control analysis. The nuclei were identified by forward- and side-scatter gating, using a 633-nm laser with a 610/20 filter. A 140-µm nozzle, a sheath pressure of 4.30 psi, an acquisition rate of up to 1200 events per second and a drop frequency of 6.05 kHz were applied (Supplementary Fig. 1). Single nuclei were sorted into two 384 well plates with each plate containing nuclei from only two separate animals that were never pooled. Each plate contained the same amount of nuclei per animal and two wells that were left blank and served as negative controls. The plates were quickly centrifuged after sorting and stored in −80°C until further processing.

**Sequencing and quality control**. cDNA libraries were produced and sequenced using a Smart-seq2 protocol[30]. Sequencing of the single-nuclei libraries was performed using Illumina HiSeq 2000. The expression values were computed as reads per kilobase of gene model and million mappable reads (RPKMs) to normalize for varying sequencing depths across sequenced nuclei and the gene lengths. The expression values of merged RefSeq and Ensembl gene annotations were computed as described by Ramsköld and colleagues[47], using uniquely aligned reads and correcting for the uniquely alignable positions using MULTo[48]. The reads were mapped and aligned to mouse genome (mm10) with the STAR aligner[49] using 2-pass alignment to have improved performance of de novo splice junction reads, filtered for only uniquely mapping reads. Average number of reads per nuclei was 682000. About 80% of the reads showed unique alignments to the mouse genome with 25% of the mapped reads aligning within exons. Only exons were included in the analysis. Parameters for quality control included uniquely mapping reads (<80%), ERCC detection (< mean −2 standard deviations), ERCC ratio (>0.1) fraction of reads mapping to exons (<10%), detected with RPKM > 1 (>3000 and < mean +2 standard deviations) and a maximum correlation to another nuclei (< 0.3).

**Bioinformatics analysis**. The obtained RPKM data was analyzed using the Seurat package 2.3.4 in R[31] to identify subpopulations followed by MAST (Model-based Analysis of Single Cell Transcriptomics)[50] to find differentially expressed genes (statistically significantly up- or downregulated) between the individual clusters. The number of statistically significantly upregulated genes detected when comparing one cluster versus the rest with a p adjusted value of <0.01 were 19 (purple cluster0), 25 (green cluster1), 50 (light-green cluster2), 141 (red cluster3), 69 (orange cluster4) and 94 (blue cluster5) for the different clusters (Supplementary Data 2), while the downregulated genes were 87, 15, 30, 149, 16 and 69 for the 6 clusters, respectively (Supplementary Data 2). Differentially expressed genes were visualized using the T-embedded stochastic neighbor embedding (t-SNE), which is a non-linear dimensionality reduction method, and violin plots or

dotplots[51]. A minimum of 3 nuclei and minimum of 200 genes was a preliminary criterion for the Seurat object. For the analysis the Seurat package was run twice following the removal of one cluster that did not meet the neuronal criteria. For the two runs variable gene selection was set for the x-low, x-high and y cut-off at 0.8, 10 and 1 for the first run and 0.6, 10 and 1 for the second run, respectively. Clustering was run with the default SNN clustering at resolution of 0.8. The number of Principal components (PCs) used for the analysis was based on a Jackstraw analysis for the first 12 PCs with a cut-off of 0.001. For the first run PCs 1-7 and 12 were used, while for the second Seurat only the first 6 PCs were included.

**Fluorescent in situ hybridization analysis using single or double probes (FISH)**. Brains of c57bl/6n anaesthetized mice (P28) were extracted and snap-frozen by rapidly immersing the tissue in ice-cold isopentane (-30°/-35°C). FISH was performed using antisense riboprobes towards the mRNA of the Pitx2, Slc17a6/Vglut2 and Gad1 genes, and of 16 genes selected from the RNASeq analysis (listed in alphabetical order) and supported by results published as single-labelings in the Allen Brain Atlas: Adcyap1, Baiap3, Cacna2d3, Calb2, Col24a1, Fgf11, Glra3, Htr2c, Kcnab3, Nmbr, Nxph1, Nxph4, Pvalb, Stxbp2, Synpr, Tac1 (Supplementary Tables 2 and 3). The FISH protocol described by Dumas and Wallén-Mackenzie (2019)[52] was implemented. Briefly, 16 µm cryo-sections were prepared, air-dried and fixed in 4% paraformaldehyde, after which they were acetylated in 0.25% acetic anhydride/100 mM triethanolamine (pH = 8) and washed in PBS. Sections were hybridized for 18 h at 65°C in 100 µl of hybridization formamide-buffer containing 1 µg/ml digoxigenin (DIG) and 1 µg/ml fluorescein-labeled probes for detection of mRNA. Sections were washed at 65°C with SSC buffers (5X and 0.2×), followed by blocking with 20% heat-inactivated fetal bovine serum and 1% blocking solution (Roche #11096176001). Sections were next incubated with horse radish peroxidase (HRP)-conjugated anti-DIG fab fragments (Sigma-Aldrich #11207733910) at a dilution of 1:2500 in phosphate buffered saline and 0.1% Tween (PBST). Signal detection was obtained using Cy3-tyramide at a dilution of 1:100. Fluorescein epitopes were then detected with HRP conjugated anti-fluorescein fab fragments (Sigma-Aldrich #11426346910) at a dilution of 1:5000 in PBST and revealed using Cy2-tyramide at a dilution of 1:250. Cy2-tyramide and Cy3-tyramide were synthesized as previously described[53]. DAPI was used for nuclear staining. All slides were scanned at 20x magnification on a NanoZoomer 2.0-HT. The Ndp2.view software (Hamamatsu) was employed for viewing the images. Anatomical areas and stereotaxic coordinates were assigned according to published mouse brain atlas[54]. Representative images were selected for presentation. S1-S3 section levels correspond to bregma −1.7/−1.82; −1.94/−2.06; −2.3/−2.46, respectively. For each region of interest, a semi-quantitative histological analysis was performed by manual counting in a minimum of 3 sections per area per brain per probe combination. Brain tissue from 2-3 mice was used for each probe combination.

**Reporter detection in tissue**. Deeply anaesthetized mice were transcardially perfused with body-temperature phosphate-buffered saline (PBS) followed by ice-cold 4% formaldehyde. Brains were dissected and post-fixed overnight. Brains were cryo-protected with 30% sucrose and cut using a cryostat at 30 µm slice thickness. mCherry was detected by the endogenous fluorescence without any additional use of an antibody. Images were captured using a NanoZoomer 20.2-HT.0 with a 20× objective and processed using the Ndp2.view software (Hamamatsu).

**Reporting Summary**. Further information on research design is available in the Nature Research Reporting Summary linked to this article.

## Data availability

All relevant data are available from the authors upon request (Å.W.M). RNA-sequencing data have been submitted to the Gene Expression Omnibus (GEO) database repository under the accession code GSE133953. Data are currently stored in resources provided by Uppsala University Multidisciplinary Center for Advanced Computational Science (UPPMAX/SNIC). All in situ hybridization data are available in the figures of the manuscript. Source data underlying plots shown in figures are provided in Supplementary Data 3.

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

## Acknowledgements

The authors thank Professors James Martin, Baylor College of Medicine (Houston, Texas, USA) for sharing the Pitx2-Cre mice and Jan Stenman (Ludwig Institute for Cancer Research, Stockholm Branch, Sweden) for the mCherryTRAP reporter mice. Konstantinos Toskas and Dr Erik Södersten are thanked for advice on nuclear isolation, and Dr Johan Holmberg is thanked for providing the lab facility for nuclear isolation (Ludwig Institute for Cancer Research, Stockholm Branch and CMB, Karolinska

Institutet, Sweden). The authors thank Jaromir Mikes for technical assistance at the FACS Facility at Science for Life Laboratory (SciLifeLab, Sweden) and Marie-Laure Niepon at the Image platform at Institute de la Vision (Paris, France) for slide scanning. The single cell transcriptome data was generated at the Eukaryotic Single-cell Genomics facility at SciLifeLab, Sweden (ESCG_agreement_0079). The computations were performed and stored on resources provided by SNIC through Uppsala University Multidisciplinary Center for Advanced Computational Science (UPPMAX) under Project b2017161 and SNIC 2017/7-252. Adriane Guillaumin and Dr Gian-Pietro Serra of the MackenzieLab are thanked for constructive comments on the manuscript. This work was supported by Uppsala University and by grants to Å.W.M from the Swedish Research Council (SMRC 2017-02039, 2014-3804), the Swedish Brain Foundation (Hjärnfonden), Parkinsonfonden, Bertil Hållsten Research Foundation, and the Zoological and Åhlén Research Foundations; post-doctoral grants to M.P (Stiftelse Olle Engkvist Byggmästare) and N.K (Hjärnfonden). Å.K.B is financially supported by the Knut and Alice Wallenberg Foundation as part of the NBIS facility at SciLifeLab, Sweden. A previous version of this manuscript, containing the snRNASeq results, was published in BioRxiv (https://doi.org/10.1101/677724). Open access funding provided by Uppsala University.

## Author contributions

Å.W.M.: Conceptualized and supervised the study, analyzed data, designed and prepared figures and wrote the manuscript; S.D.: Designed and performed in situ hybridization experiments, analyzed data and prepared figures; M.P.: Designed and performed sequencing and bioinformatics analysis, prepared figures and wrote manuscript; M.M.M-T.: Performed bioinfomatics analysis; B.V.: Prepared brain sections, analyzed data; N.K.: Prepared figures, analyzed data; Å.K.B.: Designed and supervised bioinformatics analysis. Contributions by M.M.M-T. and Å.K.B. were provided through the National Bioinformatics Infrastructure Sweden, NBIS (Project 3971), SciLifeLab, Sweden. All authors read the manuscript and approved submission.

## Competing interests

S.D. is the owner of Oramacell. The remaining authors declare no competing interests.
