## [Peer Review File · Communications Biology]

Reviewers' comments:

Reviewer #1 (Remarks to the Author):

Brief summary of the manuscript

The manuscripts presented addresses the cellular diversity of the Pitx2-positive cells with focus on the subthalamic nucleus and surrounding structures. The authors applied single nuclear sequencing and managed to identify several subtypes.

Overall impression of the work

Overall, the data and results presented seem to be valid (very good sequencing quality for nuclear sequencing) but in my opinion the manuscript has some major and minor shortcomings that need to be addressed in order to be considered publishable. I think the manuscript would benefit from increasing the cell number as well from thorough in situ confirmation of the clusters found.

Specific comments

Major Remark:

1. I do believe that the sequencing has been performed properly; however, if I understand this correctly the biological replicate is 1 which is not ok. The authors use 4 animals which are mixed together so it has to be regarded as one biological sample. Also using 2 experimental plates doesn't change that fact. I would like to see at least one repetition of this experiment using 2 mice and 1 plate to show reproducibility.
2. As seen unfortunately too often in publications the quantitative analysis is lacking. I do appreciate the authors attempt to address this point in Figure 5 using immunohistochemistry but unfortunately, the markers used here are according to their own list (suppl. Table 2) not the top hits. Additionally, the picture quality seems lacking (at least on my Pdf version) and high magnifications are missing. I would like to see better in situ validation (preferably using FISH) for the top hit (defining) genes. This would allow qualitative and quantitative evaluation of the clusters within the tissue and confirm if clusters are indeed separate (e.g. Clusters 1+2 are very similar). Due to commercially available FISH Probes/Kits from different companies, this should be very simple and fast to do for the 6 clusters (ACD-Bio or Molecular Instruments to name two easy to use Kits).
3. Due to the similarity of some clusters more validation is necessary. Does the "light green" (Fig. 3) cluster appear with more strict settings in the clustering algorithm? Can you really identify the different populations in situ?

Minor Remarks:

1. In general, the nomenclature chosen is a bit confusing. To solve this I would recommend to move the first clustering result (heatmap,...) from Fig 2 into supplementary as 1 Cluster (with

oligodendrocyte markers: Tmem125, Tmem63a, Cd9, Car14, Lpar1) is excluded.

The final cluster nomenclature should then either be 1-6 or as presented in Figure 3 using particular names that are useful should somebody else publish other cell types (e.g. GABAergic populations) in this region.

2. Clusters nomenclature should always be stated in the upcoming figures together with the color code.

3. In Fig. 3c-h legend it is stated that violin plots for 6 markers are shown but there are always 9 plots.

4. Despite realizing the authors intention I think the right panel in Fig. 6 is a bit chaotic and should be changed. Presentation could be done similar to Gioele La Manno, Daniel Gyllborg,... et al. 2016 (Cell) Fig. 5e or 6a. to make it more readily understandable.

Fig.4+6. could also benefit from in situ validation if 2-3 novel markers are depicted that display a particular spatial distribution within and/or outside the STN.

5. The quality control seems robust and reasonable but I think one should add one panel to Suppl. Fig.2 showing that both Experimental plates (or in my opinion at least 3 plates) contribute equally to the Clusters found.

6. In Suppl. Fig.3g-j, Suppl Fig. 4 and Main Fig.7 some genes seem again not very specific to one cluster. Or to say it differently, some clusters look very similar according to the presented genes. (see Major Remark 3)

7. I got a bit confused on the sequencing data:

"...Smart-seq 2 protocol 20 for full-length RNA sequencing (snRNAseq) at a depth of 200-250 million reads and with 50bp per single read (Fig. 1B, C & Supplementary Fig. 1)...."

I guess these are the total reads. However, if this is the case, why is there a range and not a total number (maybe one for plate 1 and another one for plate 2)?

8. In respect to the excluded cluster, are those Oligodendrocyte genes contaminations? Do those cells express glutamatergic or neuronal markers? If so, cells could be included in the analysis but Oligodendrocyte genes could be excluded from the clustering procedure.

Reviewer #2 (Remarks to the Author):

This manuscript describes results of single nuclei RNA sequencing of cells from mouse subthalamic nucleus (STN). The methods used are clearly described, and the data are well organized. The study is primarily descriptive, with no follow-up functional or validation analyses performed to demonstrate the utility of the dataset. There are many phrases in the manuscript which are too vague and need better explanation. The paper needs to be edited for appropriate comma usage and grammar. Figures need improvement in annotation.

1. 37: therapeutics should be singular.

2. 40: add commas after STN and projections.

3. 45: circuitry is misspelled.

4. 46: add of after symptoms.

5. 47: provide references after OCD.

6. 51: connectivity should be plural.

7. 59: add as after well.
8. 62: add a comma after However.
9. 72: delete the comma after represent.
10. 74: add a comma after moreover and replace the with our.
11. 76: add a comma after summary
12. 77: the term scope here is not appropriate.
13. 83: delete thy hyphen in mouse-line.
14. 84: reporter line is duplicated
15. 85: please check the Jackson laboratory website for use of proper mouse strain nomenclature.
16. 89: provide detail about the 4 mice. Age? Sex? Strain?
17. 92: delete and.
18. 94: delete a
19. 97: insert the after and, and delete on
20. 98: change found to expressed.
21. 99: what is meant by "variable genes"?
22. 102: add a space after 185,
23. 104: the phrase "On the first instance" is ambiguous.
24. 106: how were the clusters shown to be of glutamatergic lineage? This needs to be explained.
25. 107: which transporter? Detail is needed.
26. 109: add commas around surprisingly.
27. 110: delete possibly
28. 112: add an s to cluster, insert were after 5, and remove the d from separated.
29. 113: add a comma after plots.
30. 116: add was after dataset.
31. 119: describe what is meant by the "second run".
32. 124: replace as well as with and.
33. 127: describe the "additional cluster".
34. 130: "when looking at the number of nuclei" can be deleted.
35. 147: change when comparing to comparison of
36. 148: change it appeared to showed.
37. 151: add a comma after however.
38. 152: what is meant by "in our case"?
39. 155: add commas after cluster and Gck.
40. 156: replace that with "as these genes".
41. 161: replace "when looking at" with "in"; add comma after atlas, insert genes after these, and change detected to expressed.
42. 162: add a comma after however.
43. 163: add a comma after STN.
44. 164: insert the after be.
45. 166: ample of genes makes no sense.
46. 168: add a comma after STN, and delete and.
47. 169: "in this" is vague.
48. 173: delete the comma and space before 30.
49. 178: add a comma after clusters and delete hat.
50. 179: add a comma after indeed and delete the text up to markers.
51. 185: add a comma after indeed.
52. 186: add Atlas after Brain and replace were with are.
53. 189: add a comma after conversely
54. 191-192: the sentence is a fragment, is awkward, and is missing a parenthesis.
55. 193: exact opposite end of the STN cluster is confusing. Provide some directional orientation.
56. 196: change to similar

57. 198: add a comma after contrast.
58. 201: add a before salt.
59. 204: add a period after cSTN, delete and, insert was after Nxph4.
60. 205: add a comma after summary.
61. 206: insert comma after marker, replace using with use of.
62. 207: replace enable with help identify.
63. 208-209 delete the before MN, STN, PSTN, LHA/cSTN.
64. 213: add a comma after hypothalamus.
65. 214: elaborate on the phrase "certain degree of limitation" as this is vague.
66. 215: add a after of.
67. 216: change scope to goal.
68. 218: add a comma after identified. "general hypothalamic populations" is too vague.
69. 221: change from to using, and add a comma after importantly.
70. 223: add a comma after addition.
71. 224: add a comma after MM.
72. 225: add a comma after PSTN.
73. 227: add a comma after and and Cdh23.
74. 228: insert compared to after than, and add and after STN,
75. 229: delete but also and the.
76. 231: add a comma after hypothalamus.
77. 232: replace but also with and.
78. 239: replace shared between this and the with "expressed in the".
79. 240: replace "explain" with "represent" and nature with structure. Delete it seems that.
80. 242: delete the between by and expression.
81. 243: Add commas after axis and hand.
82. 246: change "but also to" to "and in"
83. 247: add a comma after however.
84. 251: add a comma after study.
85. 252: add a comma after moreover
86. 254: what is meant by "protein mediating"? add a comma after others
87. 255: add commas after is and however.
88. 259: add a comma after however.
89. 261: change was to were, and add a comma after rodent.
90. 262: add commas after STN and pallidum.
91. 263: delete Given and the comma after Fxyd6.
92. 264: add a comma after far.
93. 266: "could relate changes" is vague. Change its to whose.
94. 268: change "associated to" with "correlate with patterns of"
95. 269: add a comma after Fgf11+.
96. 272: delete associated with and the
97. 273: add a comma after study and change upregulated to highly expressed.
98. 274 change is to was.
99. 275: the sentence ends abruptly with by.
100. 276: add a comma after such
101. 277: add a comma after system and change in to on.
102. 278 add a comma after STN and delete in.
103. 282-283 add parentheses around "that are highly expressed in the PSTN".
104. 284-286 the sentence is long and should be broken up into 2-3 smaller ones.
105. 286-284 add parentheses around "found in the STN but not the other two clusters". Delete 'that is', and reverse the order of elucidate further.
106. 289: change to "PSTN, STN, and LHA".

107. 290: "picked up" is colloquial. Add a comma after conclude.
108. 291: "looked at" is also colloquial. Define EP.
109. 292: change addressed to addresses; add genetic before heterogeneity.
110. 294: reverse the order of explored further; delete and by
111. 337: Ensembl is misspelled.
112. 342: Define ERCC
113. All figures and legends, including the supplement, need to be edited for missing periods, and abnormalities in lettering.
114. Figure 6 cartoon is unhelpful and difficult to interpret. Suggest replacing the box with letters by a venn diagram.
115. For the graphical abstract, use of trademark/registered brands is not appropriate.

Response to reviewers - COMMSBIO-19-0948A-Z

“Single-nuclei transcriptomic analysis of the subthalamic nucleus reveals different Pitx2-positive subpopulations”

Referee expertise:

Referee #1: Single-nuclei RNA-sequencing

Referee #2: Genetics and development of subthalamic nucleus

Our answers provided in blue.

We would like to thank both reviewers for constructive comments regarding our manuscript. By addressing all comments with ample new experiments and substantial text revisions, we feel that the quality of the manuscript has improved significantly. In addition, based on the comment to include *in situ* hybridization data, we can now present several distinct gene expression patterns in the STN and surrounding brain areas. In fact, these new additions also allow the subdivision of the STN into four domains that can be distinguished by molecular markers and spatial location within the extent of the STN. We believe that the guidance from reviewers has led us to formulate a substantially improved manuscript with important conclusions that reach a new level of knowledge in the field of the STN. We are grateful for the reviewer comments that led us along the way towards these new findings. We believe that our study in the revised format will be of strong interest to the readership or the journal.

Please note that the title has changed to reflect the revised study “*Spatio-molecular domains identified in the mouse subthalamic nucleus and neighboring glutamatergic and GABAergic brain structures*”.

Below follows a detailed point by point response to all of the reviewers’ comments and how we have addressed them.

Reviewers' comments

Reviewer #1 (Remarks to the Author):

Brief summary of the manuscript

The manuscript presented addresses the cellular diversity of the Pitx2-positive cells with focus on the subthalamic nucleus and surrounding structures. The authors applied single nuclear sequencing and managed to identify several subtypes.

Overall impression of the work

Overall, the data and results presented seem to be valid (very good sequencing quality for nuclear sequencing) but in my opinion the manuscript has some major and minor shortcomings that need to be addressed in order to be considered publishable. I think the manuscript would

benefit from increasing the cell number as well from thorough *in situ* confirmation of the clusters found.

We thank you for your constructive comments on our study. While we have not increased the number of nuclei, we have clarified that 4 independent biological samples were indeed analyzed in the study. We have also performed substantial *in situ* hybridization analysis which has enabled us to go further in the conclusions about gene expression patterns in the STN and its surrounding structures.

Specific comments

Major Remark:

1. I do believe that the sequencing has been performed properly; however, if I understand this correctly the biological replicate is 1 which is not ok. The authors use 4 animals which are mixed together so it has to be regarded as one biological sample. Also using 2 experimental plates doesn't change that fact. I would like to see at least one repetition of this experiment using 2 mice and 1 plate to show reproducibility.

We would like to clarify that the sequencing data was solidly conducted from 4 biological replicates. The nuclei were sorted into 2 plates with each plate containing nuclei from only 2 animals that were never pooled. Nuclei from 4 replicates were single sorted with each plate containing same amount of nuclei per animal. This has also been clarified in the text and a schematic illustration has been added in Figure 1.

2. As seen unfortunately too often in publications the quantitative analysis is lacking. I do appreciate the authors attempt to address this point in Figure 5 using immunohistochemistry but unfortunately, the markers used here are according to their own list (suppl. Table 2) not the top hits. Additionally, the picture quality seems lacking (at least on my Pdf version) and high magnifications are missing.

I would like to see better *in situ* validation (preferably using FISH) for the top hit (defining) genes. This would allow qualitative and quantitative evaluation of the clusters within the tissue and confirm if clusters are indeed separate (e.g. Clusters 1+2 are very similar). Due to commercially available FISH Probes/Kits from different companies, this should be very simple and fast to do for the 6 clusters (ACD-Bio or Molecular Instruments to name two easy to use Kits).

We agree with comments regarding the value of adding *in situ* hybridization data for validation. The approach of showing data from Allen Brain Atlas has previously been recognized as sufficient in recent publications of RNASeq analyses (e.g. Chen *et al.* Single-Cell RNA-Seq Reveals Hypothalamic Cell Diversity, Cell Reports, 2017).

However, we do agree with the reviewer that our own high-quality *in situ* data would considerably strengthen the paper.

Therefore, substantial new experiments have been performed. The previous immunohistochemistry figure has now been removed and replaced by several figures showing

completely new FISH data. By implementing double fluorescent *in situ* analysis using 16 selected mRNAs, we now demonstrate gene expression patterns in the STN and adjacent structures. By performing substantial *in situ* hybridization analyses throughout the subthalamic area, we can identify domains within the mouse STN structure that are characterized by gene expression profiles. By comparing mRNAs to Pitx2 and to each other, we can now subdivide the mouse STN along the mediolateral and anteroposterior axes and provide a spatio-molecular map of the STN. This level of knowledge of the internal organization of the mouse STN is entirely new.

We also identify molecular distinctions between the STN and PSTN and provide markers for neuronal populations in the hypothalamus and in mammillary nuclei. These additions were made based on this comment from the reviewer, and we believe the data strengthens the manuscript considerably. With the addition of ample new data described in the Results section, however, the Discussion has much shortened to fit in the journal format.

3. Due to the similarity of some clusters more validation is necessary. Does the “light green” (Fig. 3) cluster appear with more strict settings in the clustering algorithm? Can you really identify the different populations *in situ*?

With the new FISH data, we can demonstrate which genes are expressed in the STN, and which are present in surrounding structures. Further, most genes are expressed in more than one area, but some are rather restricted. In the original submission, there was an attempt to allocate each of the six clusters to one specific brain area. However, with the FISH data, we can conclude that this approach was not optimal. Some genes did indeed match this type of anatomical identity, while others did not, both in terms of the green clusters and the other clusters. Gene expression patterns that segregate between the different Pitx2-positive brain areas have been identified.

The sequencing data guided towards the *in situ* brain mapping, and through this approach, ample new data can now be presented. The FISH data has led the way to bona fide conclusions based on *in situ* presence of each selected gene expression. We have substantially rewritten the manuscript based on these findings. Figures, Results and Discussion have been significantly updated.

Minor Remarks:

1. In general, the nomenclature chosen is a bit confusing. To solve this I would recommend to move the first clustering result (heatmap,...) from Fig 2 into supplementary as 1 Cluster (with oligodendrocyte markers: Tmem125, Tmem63a, Cd9, Car14, Lpar1) is excluded. The final cluster nomenclature should then either be 1-6 or as presented in Figure 3 using particular names that are useful should somebody else publish other cell types (e.g. GABAergic populations) in this region.

The first clustering result has now been moved into Supplementary figure 3 and the nomenclature addressed.

2. Clusters nomenclature should always be stated in the upcoming figures together with the color code.

Cluster nomenclature now follows the same manner with color and number of cluster being the same throughout figures and color codes provided in all figures.

3. In Fig. 3c-h legend it is stated that violin plots for 6 markers are shown but there are always 9 plots.

This has been corrected.

4. Despite realizing the authors intention I think the right panel in Fig. 6 is a bit chaotic and should be changed. Presentation could be done similar to Gioele La Manno, Daniel Gyllborg, ... et al. 2016 (Cell) Fig. 5e or 6a. to make it more readily understandable. Fig.4+6. could also benefit from in situ validation if 2-3 novel markers are depicted that display a particular spatial distribution within and/or outside the STN.

All figures have changed in terms of layout and information provided, and we have improved the presentation of schematic illustrations to be more readily understandable. Several new illustrations have also been made to make the data easily accessible to the reader.

5. The quality control seems robust and reasonable but I think one should add one panel to Suppl. Fig.2 showing that both Experimental plates (or in my opinion at least 3 plates) contribute equally to the Clusters found.

t-SNE showing nuclei from each experiment plate and each biological replicate have now been added in Figure 1.

6. In Suppl. Fig.3g-j, Suppl Fig. 4 and Main Fig.7 some genes seem again not very specific to one cluster. Or to say it differently, some clusters look very similar according to the presented genes. (see Major Remark 3)

Indeed, several genes are found in several clusters. The expression *in situ* in brain tissue has been thoroughly validated and results re-considered with new conclusions.

7. I got a bit confused on the sequencing data:

“...Smart-seq 2 protocol 20 for full-length RNA sequencing (snRNAseq) at a depth of 200-250 million reads and with 50bp per single read (Fig. 1B, C & Supplementary Fig. 1)....”

I guess these are the total reads. However, if this is the case, why is there a range and not a total number (maybe one for plate 1 and another one for plate 2)?

This has now been corrected to “Average number of reads per cell was 682000”.

8. In respect to the excluded cluster, are those Oligodendrocyte genes contaminations? Do those cells express glutamatergic or neuronal markers? If so, cells could be included in the analysis but Oligodendrocyte genes could be excluded from the clustering procedure.

This cluster did express glutamatergic and neuronal markers as seen in the tSNE plot. However, as those nuclei have on average a higher number of detected genes and express oligodendrocyte genes, we were worried that they may be doublets or are contaminated with fragments from oligodendrocyte cells. Given the small size of this population, we reasoned it was scientifically better to remove these 11 nuclei rather than exclude the oligodendrocyte gene list from the clustering analysis.

Reviewer #2 (Remarks to the Author):

This manuscript describes results of single nuclei RNA sequencing of cells from mouse subthalamic nucleus (STN). The methods used are clearly described, and the data are well organized. The study is primarily descriptive, with no follow-up functional or validation analyses performed to demonstrate the utility of the dataset. There are many phrases in the manuscript which are too vague and need better explanation. The paper needs to be edited for appropriate comma usage and grammar. Figures need improvement in annotation.

We thank you for your constructive comments. The manuscript has been substantially rewritten and ample new *in situ* hybridization data have been added in response to reviewer comments. This significant addition of new data allows us to make stronger conclusions about expression patterns and thus substantially improves the quality of the manuscript.

Both the results and discussion now follow our own results and interpretation of our own data closely, and avoid speculations and vague associations altogether.

All figures and legends, including the supplement, need to be edited for missing periods, and abnormalities in lettering.

Substantial editing has been performed and completely new writing has also been added in order to encompass all the new data and conclusions. The new text has been read and approved by English-speaking persons, but of course, may still contain errors.

Figure 6 cartoon is unhelpful and difficult to interpret. Suggest replacing the box with letters by a venn diagram.

We agree and this figure has been modified with a dotplot and a heatmap to summarize the seq-data and the *in situ* data, respectively. We have also added new figures with illustrations to allow the *in situ* hybridization data be easily interpreted and understood by the reader.

For the graphical abstract, use of trademark/registered brands is not appropriate.

The graphical abstract has been changed altogether to fit the new additions and conclusions of the study.

REVIEWERS' COMMENTS:

Reviewer #1 (Remarks to the Author):

Summary of the revised manuscript:

The revision has been done thoroughly and I want to thank the authors for addressing all my concerns. Especially the extensive anatomical work and the revised visual presentation are impressive. Thus, from my side there are no objections regarding the publication of this work.

I do have only minor comments/suggestions.

Comment 1:

In figure 2 some genes do have an asterix behind their name. Those are the ones tested for in situ as they fit to the provided list of probes used. Either remove the asterix or mention in the figure legend what the asterix stand for.

Comment 2:

In figure 5 potentially remove panel f+g as it does not add to the schematic presented in panel h. This would allow combining figure 5 and 6.

Comment 3:

In figure 8 the list of names in panel a) and the table in panel b) feels redundant. For me personally, the table in panel b) looks cleaner and easier to follow.

REVIEWERS' COMMENTS:

Reviewer #1 (Remarks to the Author):

Summary of the revised manuscript:

The revision has been done thoroughly and I want to thank the authors for addressing all my concerns. Especially the extensive anatomical work and the revised visual presentation are impressive. Thus, from my side there are no objections regarding the publication of this work.

Answer: Thank you.

I do have only minor comments/suggestions.

Comment 1:

In figure 2 some genes do have an asterix behind their name. Those are the ones tested for in situ as they fit to the provided list of probes used. Either remove the asterix or mention in the figure legend what the asterix stand for.

Answer: The legend has been updated accordingly.

Comment 2:

In figure 5 potentially remove panel f+g as it does not add to the schematic presented in panel h. This would allow combining figure 5 and 6.

Answer: Figure 5 has been updated to show the glutamatergic and GABAergic neurotransmitter phenotypes (derived from panel f+g) of the STN and ZI in the schematics (panel h, colored outline of STN and ZI indicates neurotransmitter phenotype) so that the schematics fully represents the data obtained. We also updated Fig 6 in a similar way by including ZI in the schematics. Taken together, we feel that the data obtained for the ZI are better presented in schematics and conclusions this way, and that the neurotransmitter phenotype of the STN and ZI is well illustrated.

Figure legend and main text also updated accordingly.

This reviewer comment helped clarify this matter and we feel that the data is better presented now.

We opt not to merge fig 5 and 6 as we prefer these to be shown in larger size for ease of interpretation of data (The subsequent Fig 7 shows many images of similar type but in smaller size).

Comment 3:

In figure 8 the list of names in panel a) and the table in panel b) feels redundant. For me personally, the table in panel b) looks cleaner and easier to follow.

Answer: Agree, the table shown in a) has been removed.